# Computationally Sufficient Reductions for Joint Multiple Matrix Estimators with Sparsity and Fusion

**Prateek Sasan**                                              *sasan.prateek@gmail.com*
*Department of Statistics*
*The Ohio State University*

**Vincent Q. Vu**                                              *vqv@stat.osu.edu*
*Department of Statistics*
*The Ohio State University*

**Reviewed on OpenReview:** *https://openreview.net/forum?id=KK9RHgSbdp*

## Abstract

We study a broad class of methods for the joint estimation of multiple sparse symmetric matrices that incorporates group and fusion penalties for borrowing strength across related matrices. This class includes extensions of popular methods for precision and covariance matrix estimation as well as PCA. We show that these methods can be unified through the lens of computational sufficiency, a recently proposed theory that can reveal hidden commonalities between seemingly disparate methods yielding both theoretical insights into the underlying optimization problems and practical advantages in terms of computational efficiency. We derive a universal screening rule that applies simultaneously to all methods in this class, allowing us to reduce the search space to block diagonal matrices. This enables streamlined algorithms that drastically reduce the runtime, making the methods far more scalable and practical for high-dimensional data analysis.

## 1 Introduction

In high-dimensional data analysis, estimating multiple sparse symmetric matrices has emerged as a crucial problem, especially when data is collected from different but related populations. These matrices are often used to represent relationships between variables, such as covariances or conditional dependencies. Applications of such estimators arise in diverse fields, including genomics, neuroscience, and finance, where understanding underlying patterns across multiple groups of observations is essential. However, as the dimensionality of the data grows, the estimation process becomes computationally challenging, particularly when the goal is to maintain sparsity in the resulting matrices.

Graphical Lasso (Yuan & Lin, 2007; Friedman et al., 2008) is a widely used method for estimating a sparse precision matrix, where the focus is on identifying conditional dependencies between variables. However, in many practical settings, there is interest in extending Graphical Lasso to jointly estimate multiple precision matrices corresponding to different but related populations. To address this, recent extensions of Graphical Lasso have incorporated additional penalties that encourage similarity and shared structure across the matrices, such as the Fused Lasso penalty (Danaher et al., 2014; Monti et al., 2014; Yang et al., 2015) and the Group Lasso penalty (Danaher et al., 2014). These extensions enable more accurate modeling of complex, multi-population data by borrowing strength across the matrices. However, they also come with significant computational costs, limiting their scalability to large datasets.

Beyond Graphical Lasso, other methods in matrix estimation include sparse PCA (principal component analysis) (d'Aspremont et al., 2004; Zou et al., 2006; Vu et al., 2013) and sparse covariance matrix estimation (Rothman, 2012; Xue et al., 2012; Liu et al., 2014). Like Graphical Lasso, these methods face similar computational challenges, especially when adapted to multi-population data.

Recognizing the computational burden of these methods, various authors have developed screening rules to enhance efficiency. Screening rules eliminate irrelevant features, reducing the search space and boosting computational efficiency. These rules, applied to both Graphical Lasso (Witten et al., 2011; Mazumder & Hastie, 2012) and its extensions for estimating $K$ related matrices (Danaher et al., 2014; Yang et al., 2015), exploit the structure of the problem by determining when the solution is a block diagonal matrix. This allows the algorithm to restrict the search space to block diagonal matrices rather than all matrices, significantly reducing the computational complexity. Screening rules thus play a crucial role in improving algorithm efficiency without sacrificing accuracy of the estimates.

A recent theory, computational sufficiency (Vu, 2018b), provides a framework for identifying shared reductions in the input data across a family of estimators. These reductions allow for recovering a solution to the original problem while revealing hidden commonalities between the estimators, as they rely on the same data simplifications. This insight not only highlights structural connections but also leads to significant algorithmic speed-ups. Vu (2018b) applied this theory to a class of sparse symmetric matrix estimators, uncovering previously unknown relationships between Graphical Lasso (Friedman et al., 2008) and sparse PCA (Vu et al., 2013). In the process, they recovered existing screening rules for Graphical Lasso and established new screening rules for the sparse PCA problem and a broader class of penalized estimation methods that share similar invariances in their loss functions. In a parallel development, Vu (2018a) applied the framework to denoising estimators with generalized lasso penalties, showing that penalized least squares is computationally minimal among estimators sharing the same penalty—with applications to fused lasso, total variation denoising on graphs, and isotonic regression.

The general computational-sufficiency framework—the projection-based theory, the group-invariance structure, and the proof techniques used to establish that a sufficient reduction exists—is due to Vu (2018b) and is what we adopt as our starting point. That work treated only the single-matrix case ($K = 1$) without a fusion penalty. The joint multiple-matrix setting does not follow as a corollary, because a fusion penalty couples the $K$ matrices and the reduction must operate on the joint data tuple rather than on each matrix in isolation. Extending the framework to this setting, deriving the penalty-specific screening graphs that make it concrete for group-lasso and generalized-lasso fusion penalties, and the resulting universal screening rule and algorithms are the contributions of this paper.

## 1.1 Contributions

In this paper, we consider a family of multiple sparse symmetric matrix estimators that operate on $K$ symmetric $d \times d$ matrices $\mathbb{X}^{(1)}, \ldots, \mathbb{X}^{(K)} \in \mathbb{R}^{d \times d}$. We organize these matrices into a tuple:

$$\mathbb{X} = \left( \mathbb{X}^{(1)}, \ldots, \mathbb{X}^{(K)} \right).$$

These matrices are usually computed from data collected from $K$ different populations. For example, $\mathbb{X}^{(k)}$ could be the sample covariance matrix of the data collected from the $k$th population, or it could be a matrix of nonparametric rank-based correlation coefficients such as the Kendall's tau or Spearman's rho (Liu et al., 2012). In any case, we take an abstract view treating these matrices as input to the estimation problem.

The main contributions of this paper are as follows:

1. We examine a general family of multiple sparse symmetric matrix estimators, including extensions of Graphical Lasso, sparse PCA, and sparse covariance matrix estimators, that utilize $L_p$ norm and Generalized Lasso penalties for borrowing strength across matrices.

2. We discover hidden commonalities among these estimators by introducing computationally sufficient reductions. This is achieved through a shared reduction utilizing a joint single-linkage thresholding operator.

3. We develop new screening rules that reduce the search space to block diagonal matrices, leading to substantial gains in computational efficiency. These rules not only recover established screening techniques for the Graphical Lasso but also uncover novel screening rules applicable to sparse PCA and other estimators. Importantly, these rules are *universal* and apply to all estimators in the family simultaneously.

It is important to emphasize that prior to this work, no screening rules existed for the general family of estimators we consider. Existing screening rules for Graphical Lasso (Witten et al., 2011; Mazumder & Hastie, 2012) apply only to the single-matrix case ($K = 1$) without fusion penalties. Extensions to joint Graphical Lasso (Danaher et al., 2014; Yang et al., 2015) handle specific fusion penalties but do not extend to sparse PCA, sparse covariance estimation, or the broader class of fusion penalties we consider. Our framework subsumes all of these as special cases: setting $K = 1$ and turning off the fusion penalty (i.e. $\lambda_2 = 0$ in (2.2)) recovers the single-matrix screening rules. Moreover, our screening rules are *universal*—they apply simultaneously to all estimators in the family, including those for which no screening rules previously existed. This universality reveals a hidden structural commonality: despite their different loss functions and intended applications, all estimators in the family depend on the data through the same reduced representation.

In the remainder of the paper, we introduce the family of multiple sparse symmetric matrix estimators in Section 2, develop a computationally sufficient reduction for the family in Section 3, discuss its algorithmic consequences in Section 4, and present experimental results in Section 5. We conclude with a discussion in Section 6. All proofs are deferred to the appendix.

**Notation**  Some of the notation we use in this paper is as follows: $\langle \cdot, \cdot \rangle$ is either a Euclidean inner product between vectors or the trace inner product between matrices; $\|\cdot\|_q$ is the $L_q$ norm of a vector, $\|\cdot\|_{1,1}$ is the elementwise $L_1$ norm of a matrix, and $\|\cdot\|_F^2 = \langle \cdot, \cdot \rangle$ is the squared Frobenius norm of a matrix. Given a tuple $\mathbb{X}$ of $K$ matrices of the same dimensions, we use the superscripted $\mathbb{X}^{(k)}$ to denote the $k$th matrix in the tuple, and subscripted $\mathbb{X}_{ij}$ to denote the $K$-vector formed by $ij$th entries of the matrices in the tuple.

## 2 Joint Estimation of Multiple Matrices

In this section, we introduce a family of estimators designed to jointly estimate $K$ sparse symmetric matrices. These estimators are formed by adding together the loss functions of separate $L_1$-penalized single matrix estimators and a penalty across the matrices that encourages structural similarities. Our family will consist of all estimators formed in this way, where the loss function satisfies a certain invariance property. But before explaining this abstraction, we first review the Graphical Lasso, its extension to the Joint Graphical Lasso, and other single matrix estimators that can be extended to multiple matrices. Then we introduce our unifying framework which encompasses all of these examples and others simultaneously.

### 2.1 From Single to Joint Graphical Lasso

The Graphical Lasso (Yuan & Lin, 2007; Friedman et al., 2008) is a popular method for estimating a precision matrix, i.e. an inverse covariance matrix, from high-dimensional data. The nonzero entries of the resulting estimate can be used to make inferences about an underlying graphical model for the data. The estimator is defined as a minimizer of the following convex loss function over positive semidefinite matrices $\theta$:

$$L(X, \theta, \lambda_1) = -\log \det(\theta) + \langle X, \theta \rangle + \lambda_1 \|\theta\|_{1,1}, \qquad (2.1)$$

where $X$ is a sample covariance matrix and $\lambda_1 > 0$, $\|\theta\|_{1,1}$ is the sum of the absolute values of the entries of $\theta$, and $\lambda_1$ is the $L_1$ penalty parameter—larger values induce more sparsity in the solution.

Graphical Lasso produces an estimate of a single precision matrix. However, in many cases, data may originate from multiple distinct distributions. For instance, a cancer researcher may have gene expression data corresponding to various subtypes of a particular cancer. In such a scenario, the set of genes (variables) would be shared, but the underlying precision matrices could differ across cancer subtypes, making it advantageous to estimate distinct precision matrices for each subtype.

Given that the samples are different subtypes of a particular cancer, we expect some degree of similarity between the precision matrices, so applying Graphical Lasso separately for each subtype may not be the most efficient use of the data. Instead, it is preferable to estimate these precision matrices jointly, modifying the optimization problem to encourage shared structure across subtypes. Several researchers have extended the Graphical Lasso framework to facilitate the joint estimation of multiple precision matrices. Given a

$K$-tuple $\mathbb{X}$ of sample covariance matrices, the Joint Graphical Lasso minimizes over a $K$-tuple of symmetric matrices $\Theta$ the composite penalized loss function:

$$\left[\sum_{k=1}^{K} L(\mathbb{X}^{(k)}, \Theta^{(k)}, \lambda_1)\right] + \lambda_2 \mathcal{P}(\Theta)\,. \tag{2.2}$$

This loss is formed by adding separate instantiations of the Graphical Lasso loss (2.1) for each matrix, and a *fusion penalty* $\mathcal{P}$ that encourages similarities among the estimated precision matrices. We give specific examples in the next section.

Several extensions of Graphical Lasso have been proposed along these lines. Danaher et al. (2014) extend Graphical Lasso by incorporating Fused Lasso penalties on a complete graph and Group Lasso penalties. Smooth Incremental Graphical Lasso (Monti et al., 2014) and Fused Multiple Graphical Lasso (Yang et al., 2015) apply Fused Lasso penalties on chain graphs. See also Pircalabelu et al. (2016); Pircalabelu & Claeskens (2020); Wu et al. (2019); Gibberd & Nelson (2014); Zhang et al. (2019); Hallac et al. (2017); Tsai et al. (2022); Qiu et al. (2016); Cai et al. (2016) for more examples.

### 2.1.1 Fusion Penalties

We consider two classes of entrywise fusion penalties in this paper. The first class consists of *Generalized Lasso* penalties (see Tibshirani & Taylor, 2011), which are defined by a full row rank matrix $B$ and take the form:

$$\mathcal{P}_B(\Theta) = \sum_{ij} \|B\Theta_{ij}\|_1\,, \tag{2.3}$$

where $\Theta_{ij}$ denotes the $K$-vector formed by collecting the $(i, j)$th entries of all matrices in the tuple, thereby enforcing structural similarity across these corresponding elements.

An example of such a penalty is the generalized Fused Lasso penalty, which takes $B$ to be an $m \times K$ oriented incidence matrix of a graph with $m$ edges on $K$ vertices, e.g. a chain graph or a complete graph. In the case of a chain graph, the Fused Lasso penalty encourages smoothness across the precision matrices, while in the case of a complete graph, it encourages similarity without regard to the ordering of the matrices.

The second class of fusion penalties are $L_q$ *Group Lasso* penalties (Yuan & Lin, 2006), which are given by,

$$\mathcal{P}_{\ell_q}(\Theta) = \sum_{ij} \|\Theta_{ij}\|_q\,, \tag{2.4}$$

where $q \in [1, \infty]$. Common choices are $q = 2$, which gives the standard Group Lasso penalty, and $q = \infty$, which encourages entries in a group to have similar magnitudes.

## 2.2 Single Matrix Estimators

In the previous section, we saw how Graphical Lasso can be extended to the joint estimation of $K$ related precision matrices by combining separate single Graphical Lasso problems with a fusion penalty. Here we show more examples of single matrix estimators that can be extended in a similar way: Sparse PCA, sparse covariance matrices, and the parameter matrix of an Ising model.

We consider sparse symmetric matrix estimators based on the $L_1$ penalized loss

$$L_A(X, \theta, \lambda_1) = A(\theta) - \langle X, \theta \rangle + \lambda_1 \|\theta\|_{1,1}\,, \tag{2.5}$$

where $A$ is a proper closed convex function possibly taking extended real values, called the *generator*, and $X$ is a symmetric matrix, e.g. a sample covariance matrix. The loss function is a generalization of the penalized log-likelihood of an exponential family with $A$ playing the role of a log-partition function.

**Graphical Lasso**   Graphical Lasso (2.1) is a special case of (2.5) with

$$A(\theta) = \begin{cases} -\log\det(-\theta) & \text{if } -\theta \succeq 0 \\ +\infty & \text{otherwise.} \end{cases} \tag{2.6}$$

Note that here $\theta$ is the *negative* precision matrix, i.e. the $\theta$ in (2.1) is actually $-\theta$ in (2.5) with this $A$.

**Sparse PCA**   D'Aspremont et al. (2004) introduced a convex relaxation of the sparse PCA problem for a single eigenvector. This was later extended to the case of $r < d$ eigenvectors by Vu et al. (2013). They recognized that PCA could be formulated in terms of rank-$r$ projection matrices and considered the convex hull of these matrices, the *Fantope*:

$$\mathcal{F}^r = \{\theta \in \mathbb{R}^{d \times d} \mid \langle I, \theta \rangle = r,\ 0 \preceq \theta \preceq I\}.$$

This is the set of symmetric matrices with eigenvalues between 0 and 1 and whose trace is $r$. Then the convex relaxation of the sparse PCA problem is formulated as (2.5) with the generator $A$ taken to be the convex indicator function of the Fantope, i.e.

$$A(\theta) = \begin{cases} 0 & \text{if } \theta \in \mathcal{F}^r \\ +\infty & \text{otherwise.} \end{cases}$$

**Sparse Covariance**   Unlike the Graphical Lasso, estimating a covariance matrix by combining a Gaussian log-likelihood with an $L_1$ penalty leads to a challenging nonconvex problem (Bien & Tibshirani, 2011). Simple methods based on thresholding the sample covariance matrix have been proposed (Rothman et al., 2009) and can be viewed as penalized least squares estimators, but they do not guarantee positive definiteness of the estimate. To address this, Rothman (2012) proposed a method that introduces a log-determinant barrier that forces the estimator to be positive definite. Their estimator takes the form of (2.5) with

$$A(\theta) = \begin{cases} \frac{1}{2}\|\theta\|_F^2 - \tau\log\det(\theta) & \text{if } \theta \succeq 0 \\ +\infty & \text{otherwise,} \end{cases}$$

where $\tau > 0$ is a small positive constant.

Xue et al. (2012) pointed out that the log-determinant penalty can be problematic when the smallest eigenvalue of the true covariance matrix is not bounded away from zero. They instead proposed an estimator that takes the form of (2.5) with

$$A(\theta) = \begin{cases} \frac{1}{2}\|\theta\|_F^2 & \text{if } \theta \succeq \epsilon I \\ +\infty & \text{otherwise} \end{cases},$$

where $\epsilon > 0$ is a lower bound on the smallest eigenvalue of the true covariance matrix. This estimator was also proposed by Liu et al. (2014).

**Sparse Ising Model**   Our last example is the Ising model, which is a popular Markov Random Field model for multivariate binary data (Ravikumar et al., 2010). Given $d$-vectors with $-1$ and $+1$ entries, the Ising model is a discrete analog of the Gaussian graphical model. The $L_1$-penalized maximum likelihood estimator of the interaction matrix $\theta$ in such a model takes the form of (2.5) with the generator $A$ given by the log-partition function of the Ising model:

$$A(\theta) = \log\left\{ \sum_{u \in \{-1,+1\}^d} \exp(\langle uu^\top, \theta \rangle) \right\}.$$

Solving the resulting optimization problem is challenging, and extending this model to multiple matrices is even more so, but we will show later that the reductions that we propose can be applied to this problem as well.

### 2.3 Families of Joint Estimators with Invariant Generators

In the previous sections, we introduced (i) several single-matrix estimators and their associated generator functions, and (ii) two classes of entrywise fusion penalties. We now combine these building blocks into a unified framework that encompasses all of those methods as special cases. This subsection defines the family of joint estimators that we will analyze throughout the rest of the paper.

Our starting point is the general recipe used in (2.2). For any generator $A$ and fusion penalty $\mathcal{P}$, we consider the composite penalized loss

$$\mathcal{L}_{A,\mathcal{P}}(\mathbb{X}, \Theta, \lambda_1, \lambda_2) = \sum_{k=1}^{K} L_A(\mathbb{X}^{(k)}, \Theta^{(k)}, \lambda_1) + \lambda_2 \mathcal{P}(\Theta). \tag{2.7}$$

The first term in the loss,

$$\sum_{k=1}^{K} L_A(\mathbb{X}^{(k)}, \Theta^{(k)}, \lambda_1) \tag{2.8}$$

is simply the sum of the $K$ single-matrix losses defined by the generator $A$, whereas the second term, $\lambda_2 \mathcal{P}(\Theta)$, couples the $K$ matrices and induces similarity across their corresponding entries through the chosen fusion penalty. This construction yields a broad class of joint multiple–matrix estimators in which $\lambda_1$ controls sparsity, and $\lambda_2$ controls similarity across matrices (through the fusion penalty $\mathcal{P}$).

The loss is proper, closed, and convex in $\Theta$, so it always admits at least one minimizer. Because the minimizer may not be unique, our estimators return the set of minimizers rather than a single point. Formally, this is expressed as a map

$$T_A : (\mathbb{R}^{d \times d})^K \to 2^{(\mathbb{R}^{d \times d})^K}$$

where $(\mathbb{R}^{d \times d})^K$ is the space of $K$ symmetric matrices and $2^{(\mathbb{R}^{d \times d})^K}$ is its power set, so the output is a set of solutions rather than a single matrix tuple.

**Definition 2.1.** Let $A : \mathbb{R}^{d \times d} \to \mathbb{R} \cup \{+\infty\}$ be a closed convex function defined on symmetric matrices and $\mathcal{P}$ be either the Generalized Lasso penalty (2.3) or the Group Lasso penalty (2.4). The family of estimators with *diagonal sign conjugation* invariant generators, denoted $\mathcal{M}_{\mathcal{P}}(\lambda_1, \lambda_2)$, consists of the set-valued estimators

$$T_A : (\mathbb{R}^{d \times d})^K \to 2^{(\mathbb{R}^{d \times d})^K}$$

defined by

$$T_A(\mathbb{X}) = \arg\min_{\Theta} \mathcal{L}_{A,\mathcal{P}}(\mathbb{X}, \Theta, \lambda_1, \lambda_2), \tag{2.9}$$

as $A$ ranges over all closed convex functions that are invariant under conjugation by diagonal sign matrices, i.e. $A(D\theta D) = A(\theta)$ for any diagonal matrix with entries $+1$ or $-1$ along its diagonal.

This definition formalizes the full family of joint estimators. It is straightforward to verify that the diagonal sign conjugation invariance property is satisfied by *all* generators of the single matrix estimators that we have introduced in the previous section. The invariance condition is essential for the unified reduction results to follow. In particular, we will show that for a fixed fusion penalty $\mathcal{P}$ and penalty values $\lambda_1$ and $\lambda_2$, there is a shared reduction that can be applied to all estimators in the family $\mathcal{M}_{\mathcal{P}}(\lambda_1, \lambda_2)$.

## 3 Computationally Sufficient Reductions

Computational sufficiency (Vu, 2018b) provides a theoretical framework to identify a common reduction in the input shared by all estimators in a family. In this section, we first introduce computational sufficiency and then present a computationally sufficient reduction for our family of estimators in Definition 2.1 and discuss its consequences.

### 3.1 Computational Sufficiency

The basic idea of computational sufficiency is to extend the likelihood interpretation of the classical notion of statistical sufficiency to families of estimators without requiring a formal statistical model. We would like to say that all estimators in a family share a common reduction $R$ in the sense that the estimators depend on the data $x$ only through $R(x)$. One difficulty with this is that for estimators based on optimization, there may not be a unique solution for a given $x$, so that the choice of solution may introduce a dependency beyond the data. So instead we require that if there is a solution, then *at least one* solution depends only on $R(x)$.

**Definition 3.1.** (Vu, 2018b) Let $\mathcal{X}$ be an input space, $\mathcal{T}$ a parameter space, and $\mathcal{M}$ a collection of set-valued estimators $T : \mathcal{X} \to 2^{\mathcal{T}}$ (each $T(x) \subseteq \mathcal{T}$ is the set of outputs—possibly more than one when the estimator is defined as the argmin of a non-strictly-convex objective). A function $R : \mathcal{X} \to \mathcal{Y}$, valued in some space $\mathcal{Y}$, is *computationally sufficient* for $\mathcal{M}$ if for each $T \in \mathcal{M}$, there exists a set-valued function $f_T : \mathcal{Y} \to 2^{\mathcal{T}}$ such that

$$f_T(R(x)) \subseteq T(x) \quad \forall\, x \in \mathcal{X}$$

and $f_T(R(x)) \neq \emptyset$ whenever $T(x) \neq \emptyset$.

For the family $\mathcal{M}_{\mathcal{P}}(\lambda_1, \lambda_2)$ studied in this paper, both the input space $\mathcal{X}$ and the parameter space $\mathcal{T}$ equal $(\mathbb{R}^{d \times d})^K$, i.e. $K$-tuples of symmetric $d \times d$ matrices.

The function $R$ in Definition 3.1 is a reduction if it is not injective. In that sense, the definition says that if a computationally sufficient reduction exists, then the estimators in the family $\mathcal{M}$ depend on the data only through the reduction. When such a reduction is known concretely, it can give insight into similarities between the estimators in the family, and can also yield insight into the structure of the solution space.

In addition to these benefits, computational sufficiency ensures that applying the reduction $R$ does not alter the solution of any estimator in the family. If an estimator $T(x)$ has a unique solution, then the output obtained by solving the reduced problem—namely $f_T(R(x))$—is exactly the same solution: no information relevant to the minimizer is lost. If $T(x)$ is set-valued, then the definition guarantees that the reduced problem still returns a solution that is one of the valid minimizers of the original full-data problem. Thus, whether the estimator yields a unique solution or a full solution set, the reduction introduces no approximation error whatsoever.

Moreover, this equivalence is entirely algorithm-agnostic. The definition makes no assumptions about how the estimators in $\mathcal{M}$ are computed; any optimization method applied to the reduced problem will return solutions that remain valid for the original problem. Thus, computational sufficiency is a purely statistical and geometric property of the estimators, not a property of a particular algorithm or implementation.

When a concrete reduction exists, it therefore provides both conceptual insight—revealing what aspects of the data truly matter for all estimators in the family—and practical benefits, since one may work with a lower-dimensional or simplified representation of the data without altering the resulting solution set.

In the context of our family of estimators in Definition 2.1, we will (1) construct an explicit reduction $R$ that is computationally sufficient for $\mathcal{M}_{\mathcal{P}}(\lambda_1, \lambda_2)$ and (2) show that it can be used to simplify the search for solutions of (2.7) simultaneously across all estimators in $\mathcal{M}_{\mathcal{P}}(\lambda_1, \lambda_2)$.

### 3.2 Joint Single-Linkage Thresholding is Computationally Sufficient

For our family of estimators $\mathcal{M}_{\mathcal{P}}(\lambda_1, \lambda_2)$, we will use a reduction based on *joint single-linkage thresholding*, an operator on $K$-tuples of $d \times d$ matrices that is defined by a graph on the index set $\{1, \dots, d\}$. We define the operator on a generic graph here, and specify the graph appropriate for our family in Section 3.3.

**Definition 3.2.** A *joint single-linkage thresholding* is an operator on $K$-tuples of $d \times d$ matrices defined by a graph $\mathcal{G}$ on the vertex set $\{1, \dots, d\}$. Specifically, the *single-linkage clustering matrix* of $\mathcal{G}$ is the binary matrix $\mathrm{SLC}(\mathcal{G}) \in \{0, 1\}^{d \times d}$ with entries

$$[\mathrm{SLC}(\mathcal{G})]_{ij} = \begin{cases} 1 & \text{if } i = j \text{ or } i \sim_{\mathcal{G}} j, \\ 0 & \text{otherwise,} \end{cases}$$

where $i \sim_{\mathcal{G}} j$ means $i$ and $j$ lie in the same connected component of $\mathcal{G}$. The associated *joint single-linkage thresholding operator* $\mathrm{SLT}(\mathcal{G})$ takes a $K$-tuple $\mathbb{Z} = (\mathbb{Z}^{(1)}, \ldots, \mathbb{Z}^{(K)})$ to its entrywise (Hadamard) product with $\mathrm{SLC}(\mathcal{G})$ in each coordinate:

$$\mathrm{SLT}(\mathcal{G})(\mathbb{Z}) = \left(\mathrm{SLC}(\mathcal{G}) \circ \mathbb{Z}^{(1)}, \ldots, \mathrm{SLC}(\mathcal{G}) \circ \mathbb{Z}^{(K)}\right).$$

The operator $\mathrm{SLT}(\mathcal{G})$ is the orthogonal projection (with respect to the entrywise inner product) onto the linear subspace

$$\mathcal{B}_{\mathcal{G}} := \left\{ \mathbb{Z} \in (\mathbb{R}^{d \times d})^K \mid \mathrm{SLT}(\mathcal{G})(\mathbb{Z}) = \mathbb{Z} \right\},$$

i.e. the subspace of $K$-tuples whose matrices are block diagonal (after a common permutation of $\{1, \ldots, d\}$) with blocks indexed by the connected components of $\mathcal{G}$. In particular, $\mathrm{SLT}(\mathcal{G}) \circ \mathrm{SLT}(\mathcal{G}) = \mathrm{SLT}(\mathcal{G})$.

In Section 3.3 we construct a *penalty-specific graph* $\mathcal{G}_{\mathcal{P}}(\mathbb{X}, \lambda_1, \lambda_2)$ on $\{1, \ldots, d\}$ whose edges identify the coordinate pairs that the fusion penalty cannot force to fuse, given the data $\mathbb{X}$ and parameters $\lambda_1, \lambda_2$. To streamline notation we abbreviate

$$\mathrm{SLT}_{\mathcal{P}}(\mathbb{X}, \lambda_1, \lambda_2) := \mathrm{SLT}(\mathcal{G}_{\mathcal{P}}(\mathbb{X}, \lambda_1, \lambda_2)), \quad \mathcal{B}_{\mathcal{P}}(\mathbb{X}, \lambda_1, \lambda_2) := \mathcal{B}_{\mathcal{G}_{\mathcal{P}}(\mathbb{X}, \lambda_1, \lambda_2)},$$

and likewise for $\mathrm{SLC}_{\mathcal{P}}(\mathbb{X}, \lambda_1, \lambda_2)$.

Figure 1 illustrates the joint single-linkage thresholding at $d = 5$, $K = 3$ on a small example: a graph $\mathcal{G}$ with two connected components $\{1, 2\}$ and $\{3, 4, 5\}$ produces the block-diagonal mask $W = \mathrm{SLC}(\mathcal{G})$, and $\mathrm{SLT}(\mathcal{G})$ acts on the tuple $\mathbb{X}$ by zeroing every off-block entry of every slice.

**Theorem 3.3.** *Let $\mathcal{G}_{\mathcal{P}}(\mathbb{X}, \lambda_1, \lambda_2)$ be the penalty-specific graph from Section 3.3, and let $R(\mathbb{X})$ be the corresponding joint single-linkage thresholding of the data, $R(\mathbb{X}) = \mathrm{SLT}_{\mathcal{P}}(\mathbb{X}, \lambda_1, \lambda_2)(\mathbb{X})$. Then for each $T \in \mathcal{M}_{\mathcal{P}}(\lambda_1, \lambda_2)$ and $\widehat{\Theta} \in T(R(\mathbb{X}))$,*

$$\mathrm{SLT}_{\mathcal{P}}(\mathbb{X}, \lambda_1, \lambda_2)(\widehat{\Theta}) \in T(\mathbb{X}). \tag{3.1}$$

*Therefore, taking*

$$f_T(R(\mathbb{X})) = \{\mathrm{SLT}_{\mathcal{P}}(\mathbb{X}, \lambda_1, \lambda_2)(\widehat{\Theta}) \mid \widehat{\Theta} \in T(R(\mathbb{X}))\},$$

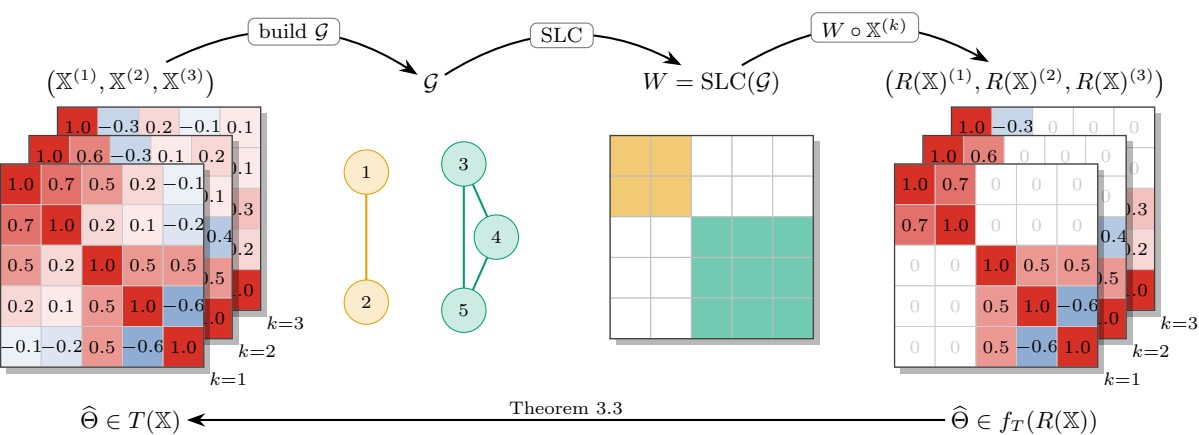

Figure 1: The joint single-linkage thresholding operator at $d = 5$, $K = 3$, with two connected components in $\mathcal{G}$ shown in orange ($\{1, 2\}$) and teal ($\{3, 4, 5\}$). *Top:* input tuple $(\mathbb{X}^{(1)}, \mathbb{X}^{(2)}, \mathbb{X}^{(3)})$, graph $\mathcal{G}$, the binary mask $W = \mathrm{SLC}(\mathcal{G})$, and the reduced data $R(\mathbb{X})^{(k)} = W \circ \mathbb{X}^{(k)}$. The graph and mask are computed from the tuple and shared across all $K$ slices; rows and columns have been labelled so that the components are contiguous. *Bottom:* Theorem 3.3 guarantees that if $\mathcal{G}$ is the penalty-specific graph from Section 3.3, then any $\widehat{\Theta} \in f_T(R(\mathbb{X}))$ obtained on the reduced data is also a valid solution on the original data, $\widehat{\Theta} \in T(\mathbb{X})$.

*the reduction R is computationally sufficient (in the sense of Definition 3.1) for the family $\mathcal{M}_{\mathcal{P}}(\lambda_1, \lambda_2)$ defined in Definition 2.1.*

The proof is given in the appendix.

Theorem 3.3 formalizes a key structural simplification shared by all estimators in $\mathcal{M}_{\mathcal{P}}(\lambda_1, \lambda_2)$. Although these procedures may differ in how they solve their respective optimization problems, they all operate on data only through the same reduced form $R(\mathbb{X})$ and any solution produced by an estimator on the reduced data can be "lifted back" to a valid solution on the original data simply by reapplying the same thresholding operator. This has several immediate implications.

**Common preliminary computation** Every estimator begins with the same reduction step, regardless of how the rest of the procedure is formulated. The penalty-specific graph $\mathcal{G}_{\mathcal{P}}(\mathbb{X}, \lambda_1, \lambda_2)$ encodes all pairwise relationships across the $K$ matrices that are strong enough—relative to $\lambda_1$ and $\lambda_2$ —to force coordinates to behave as a single fused block. Constructing it therefore identifies the essential grouping structure that any valid solution must respect.

**Shared reduced search space** By Equation (3.1), every estimator in $\mathcal{M}_{\mathcal{P}}(\lambda_1, \lambda_2)$ has at least one minimizer in the block-diagonal subspace $\mathcal{B}_{\mathcal{P}}(\mathbb{X}, \lambda_1, \lambda_2)$; hence we may intersect the feasible set of (2.7) with this subspace and still recover a valid minimizer of the original problem (although when the minimizer is not unique, some minimizers may lie outside $\mathcal{B}_{\mathcal{P}}(\mathbb{X}, \lambda_1, \lambda_2)$). The block structure is thus an intrinsic property of the fusion penalty and the data, not of a specific algorithm. Corollary 3.4 below formalizes the algorithmic consequence: any first-order method[1] applied to the reduced problem operates entirely on block-diagonal iterates, so the per-iteration cost depends only on the size of the largest block.

**Corollary 3.4** (First-order methods on the reduced problem). *Let $C_1, \ldots, C_m$ be the connected components of $\mathcal{G}_{\mathcal{P}}(\mathbb{X}, \lambda_1, \lambda_2)$, and write $\mathcal{B} = \mathcal{B}_{\mathcal{P}}(\mathbb{X}, \lambda_1, \lambda_2)$ for the corresponding block-diagonal subspace. By Equation (3.1), the reduced problem*

$$\arg\min_{\Theta \in \mathcal{B}} \mathcal{L}_{A,\mathcal{P}}\big(R(\mathbb{X}), \Theta, \lambda_1, \lambda_2\big) \tag{3.2}$$

*contains at least one minimizer of the original problem on $\mathbb{X}$. The gradient, subgradient, and proximal operator of the objective in (3.2) all take values in $\mathcal{B}$. Consequently, any first-order method—gradient descent, proximal gradient, FISTA, ADMM, etc.—initialized in $\mathcal{B}$ remains in $\mathcal{B}$ throughout iteration, operating only on the $\sum_b |C_b|^2 \leq d^2$ within-block entries and ignoring the off-block entries that the reduction has identified as zero.*

The proof is given in the appendix.

**Computational benefits** The savings guaranteed by Corollary 3.4 are largest for operations whose cost grows superlinearly in dimension. Eigendecompositions, matrix inversions, and log-determinant evaluations —the dominant per-iteration costs for the generators in Section 2—all decompose into blockwise analogs. When blocks are small or many coordinates fuse together, this can reduce runtime by orders of magnitude.

**Dependence on $\lambda_1$ and $\lambda_2$** The amount of reduction is data-adaptive. If all entries satisfy $\mathbb{X}_{ij}^{(k)} \leq \lambda_1$, the entire family collapses to estimators over diagonal matrices: the penalty forces every off-diagonal entry to fuse to zero. If $\lambda_1 = \lambda_2 = 0$, then the operator introduces no constraints and we recover the full original search space. Between these extremes, the practical speedup is governed by the component sizes $|C_1|, \ldots, |C_m|$ of the screening graph: by Corollary 3.4, the per-iteration cost scales with $\sum_b |C_b|^2$, which is at most $d^2$ and is much smaller when the graph fragments. The largest gains therefore arise at moderate-to-strong regularization—the regime in which the family $\mathcal{M}_{\mathcal{P}}(\lambda_1, \lambda_2)$ is designed to be used and where many off-diagonals of $\mathbb{X}$ fall below $\lambda_1$ or are aggregated below $\lambda_2$ by the fusion structure. At the opposite end the screening graph has a single connected component and the reduction is a no-op rather than a slowdown.

---

[1]The same argument extends to Newton, quasi-Newton, and interior-point methods, since restrictions of self-adjoint operators (and their inverses) to $\mathcal{B}$ preserve $\mathcal{B}$.

### 3.3 Penalty-Specific Graphs

In the previous section, we presented the main theorem and discussed its implications. An important detail left to address is the structure of the graph $\mathcal{G}$ depending on $(\mathcal{P}, \mathbb{X}, \lambda_1, \lambda_2)$ that is required by the joint single-linkage thresholding operator.

Let $V = \{1, \ldots, d\}$ and $E \subseteq V \times V$ denote the vertex and edge sets of the graph $\mathcal{G}$, respectively. Our objective is to determine the edge set $E$ for each specific fusion penalty. We focus on identifying conditions under which $(i, j) \notin E$. These conditions are derived as part of the process to prove the theorem and further details are included in the appendix. In the following, let $\mathbb{X}_{ij}^{(k)}$ denote the $(i, j)$ entry of the $k$-th matrix in the input tuple $\mathbb{X}$.

#### 3.3.1 Group Lasso Penalty

For the Group Lasso penalty $\mathcal{P}_{\ell_q}$, the construction of the penalty-specific graph is simple. Let $r \in [1, \infty]$ such that $1/q + 1/r = 1$, i.e. so that $\ell_r$ is dual to $\ell_q$. Then

$$(i, j) \notin E \iff \sum_{k=1}^{K} \max(|\mathbb{X}_{ij}^{(k)}| - \lambda_1, 0)^r \leq \lambda_2^r.$$

For example, if $q = \infty$, then $r = 1$ and the condition simplifies to

$$(i, j) \notin E \iff \sum_{k=1}^{K} \max(|\mathbb{X}_{ij}^{(k)}| - \lambda_1, 0) \leq \lambda_2.$$

#### 3.3.2 Generalized Lasso Penalty

For the Generalized Lasso penalty $\mathcal{P}_B$, the condition for elimination of an edge $(i, j)$ involves verifying a set of linear inequalities in the input tuple $\mathbb{X}$. Let

$$g_{ij}(y) = \langle \mathbb{X}_{ij}, y \rangle - \lambda_1 \|y\|_1 - \lambda_2 \|By\|_1.$$

Then

$$(i, j) \notin E \iff g_{ij}(y) \leq 0 \text{ for all } y \in \mathbb{R}^K.$$

On the surface, this condition seems complex. However, using similar ideas from Yang et al. (2015) we can reduce the condition to verifying a finite set of linear inequalities. Note that $g_{ij}$ is piecewise linear, because it is the pointwise maximum of linear functions, and that its regions of linearity form a finite partition of $\mathbb{R}^K$ into polyhedral cones. Each of these cones is generated by the conic hull of a finite set of points, and so it suffices to verify the condition at these points. Notably, these points depend only on the matrix $B$ and not on the input $\mathbb{X}$ nor the penalty parameters $\lambda_1, \lambda_2$. So the process of finding these points can be done once for a given $B$. We provide more details in the appendix.

## 4 Algorithmic Consequences

In the previous sections, we established computationally sufficient reductions for the family of estimators with diagonal sign–conjugation invariant generators (Definition 2.1). These results hold independently of how any particular estimator is computed: the reduction introduces no approximation error, does not rely on algorithmic details, and preserves the full solution set for every estimator in the family.

In this section, we illustrate how these mathematically exact reductions can be used to improve practical algorithms. The goal here is not to propose new optimization methods or to claim that our theoretical guarantees depend on these implementations. Rather, we show how the block structure induced by the reduction naturally leads to simpler subproblems, reduced dimensionality, and faster runtimes in concrete settings.

We begin by outlining the general strategy for incorporating the reduction into existing optimization routines. We then present two representative examples based on extensions of Sparse PCA via Fantope Projection: one with a Group Lasso ($L_2$) penalty and another with a Fused Lasso penalty. These examples demonstrate how the reduction can be integrated into practice; the theoretical results themselves hold for any algorithm that computes an estimator in the family.

### 4.1 General Recipe

Fix a family of estimators $\mathcal{M}_{\mathcal{P}}(\lambda_1, \lambda_2)$ as in Definition 2.1. The general recipe is as follows:

1. Construct the penalty-specific graph $\mathcal{G}$ dependent on the data tuple $\mathbb{X}$, fusion penalty $\mathcal{P}$, and penalty parameters $\lambda_1, \lambda_2$.

2. Find the connected components of $\mathcal{G}$.

3. Compute the computationally sufficient statistic $R(\mathbb{X})$ by zeroing out any $\mathbb{X}_{ij}$ such that $i$ and $j$ are in different connected components.

4. Then for any $T \in \mathcal{M}_{\mathcal{P}}(\lambda_1, \lambda_2)$, we run an algorithm for $T$ using $R(\mathbb{X})$ as input and restricting the feasible set to those tuples $\Theta$ where $\Theta_{ij} = 0$ whenever $i$ and $j$ are in different connected components.

Details of implementation matter, of course, but we note that the first three steps require at worst linear time in the size of $\mathbb{X}$, i.e. $\mathcal{O}(d^2 K)$ operations. It is the fourth step that is estimator-specific and will require more work to implement. For the Graphical Lasso, the computation decouples completely across the connected components; for other estimators in the family that may not be the case, but the reduced feasible set can still be exploited to reduce the computational cost of the algorithm. We illustrate this with an enhancement of an ADMM algorithm for Joint Sparse PCA.

### 4.2 Example: Joint Sparse PCA

Consider an extension of Sparse PCA via Fantope Projection (Vu et al., 2013) to the case where we have data from $K$ distinct but related groups. We add a Group Lasso penalty to borrow strength between the $K$ different groups. The optimization problem, following Section 2 is given by

$$\arg\min_{\Theta} \left[ \sum_{k=1}^{K} \iota_{\mathcal{F}^r}(\Theta^{(k)}) - \langle \mathbb{X}^{(k)}, \Theta^{(k)} \rangle + \lambda_1 \|\Theta^{(k)}\|_{1,1} \right] + \lambda_2 \mathcal{P}_{\ell_2}(\Theta).$$

where $\iota_{\mathcal{F}^r}$ is the convex indicator of the trace-$r$ Fantope.

Drawing inspiration from algorithms for Sparse PCA via Fantope Projection (Vu et al., 2013) and Group Graphical Lasso (Danaher et al., 2014), we adopt the ADMM framework (Boyd et al., 2011) to solve the optimization problem. In general, the ADMM algorithm can be applied to (2.9) by splitting the objective (2.7) into two parts: one for the generator and linear term, and the other for the sparsity and fusion penalties. Then each iteration of the algorithm steps through computing a proximal operator associated with the generator, computing a proximal operator associated with the sparsity and fusion penalties, and updating the dual variables associated with the split.

For Joint Sparse PCA, the major steps become:

1. **Fantope projection**: This step involves an eigendecomposition and transformation of the eigenvalues.

2. **Generalized thresholding**: This step involves solving the proximal operator for the sparse Group Lasso penalty which involves applying elementwise soft-thresholding followed by Group Lasso thresholding.

3. Update the dual variables.

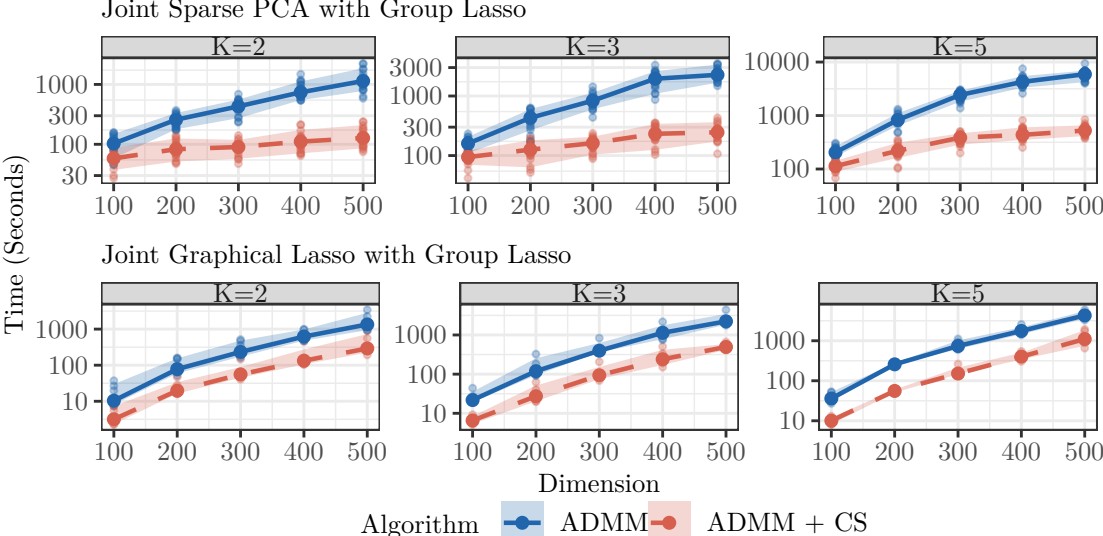

Figure 2: Total wall-clock time to compute a full regularization path of 100 penalty configurations (5 values of $\lambda_1 \times 20$ values of $\lambda_2$) for joint Sparse PCA and joint Graphical Lasso (both with Group Lasso penalty), comparing standard ADMM to its CS-accelerated variant; the full path is the standard workflow when tuning by cross-validation. Facets indicate the number of matrices $K$, and the $x$-axis is the matrix dimension $d$. Lines show the median runtime over 20 independent runs for each $(K, d)$ setting; the shaded region shows the interquartile range (25th–75th percentile). The $y$-axis is on a log scale. Across all settings, ADMM with the CS reduction reduces total path runtime by roughly 75%, with the advantage growing as $d$ increases.

The major bottleneck of the iteration is the Fantope projection step. Since we know from $R(\mathbb{X})$ the block diagonal structure of the solution, we can reduce the eigendecomposition to be block-wise; this allows the eigen decomposition to be computed on each block independently, significantly reducing the computational cost. After the eigendecomposition, we recouple the blocks by a transformation of the union of the eigenvalues from all of the blocks. Details of the algorithm are given in the appendix.

If instead of a Group Lasso penalty, we had a Fused Lasso penalty, i.e. with $\mathcal{P}_B$ for an oriented incidence matrix $B$, the algorithm would be similar, with the only difference being that in the second step, we would replace the proximal operator for a Sparse Group Lasso penalty with the proximal operator for a Sparse Fused Lasso penalty. In this case Hoefling (2010) provides an efficient algorithm for computing the proximal operator, and the bottleneck of the algorithm in the Fantope projection step can be mitigated in the same way as before.

## 5    Numerical Experiments

We empirically validate the computational benefits of the computationally sufficient (CS) reduction in two complementary regimes. Section 5.1 measures the *total* wall-clock time to compute a full regularization path—the natural unit for cross-validation workflows—under two different first-order solvers, demonstrating that the speedup is solver-agnostic. Section 5.2 then examines how the tuning parameters $(\lambda_1, \lambda_2)$ affect the time to compute a single solution. Full details of the experimental setup, including the computational specifications and a link to the code, are provided in Section 4 of the appendix.

### 5.1    Full Regularization Path

For each estimator we measure total wall-clock time to compute a path of 100 penalty configurations (5 values of $\lambda_1 \times 20$ values of $\lambda_2$), sweeping $K \in \{2, 3, 5\}$ and $d \in \{100, 200, 300, 400, 500\}$ with 20 independent replicates per $(K, d)$ setting.

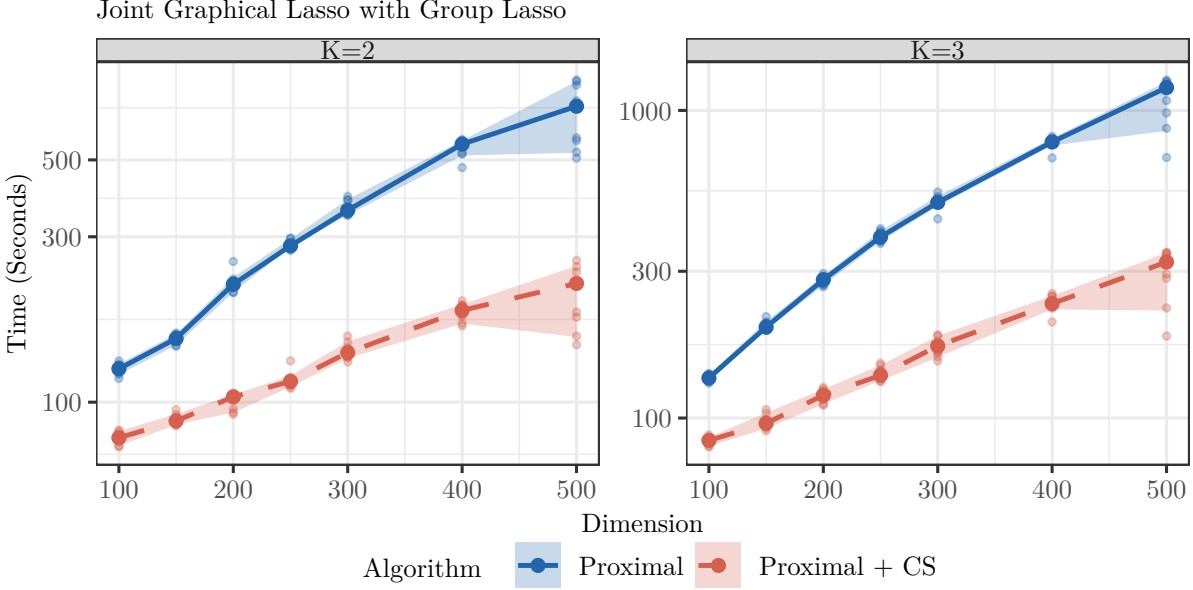

Figure 3: Total wall-clock time to compute a full regularization path of 100 penalty configurations (5 values of $\lambda_1 \times 20$ values of $\lambda_2$) for joint Graphical Lasso with Group Lasso penalty, comparing a standard proximal-gradient algorithm to its CS-accelerated variant. Facets indicate the number of matrices $K$, and the $x$-axis is the matrix dimension $d$. Lines show the median runtime over 20 independent runs for each $(K, d)$ setting; the shaded region shows the interquartile range (25th–75th percentile). The $y$-axis is on a log scale.

The choice of solver is not essential to the reduction: Corollary 3.4 guarantees that *any* first-order method applied to the reduced problem operates entirely on block-diagonal iterates. We use ADMM as the primary solver for both estimators. For Joint Sparse PCA, ADMM is the natural choice because the proximal operator combining the Fantope indicator with the $\ell_1$ and fusion terms has no closed form; splitting the Fantope projection from the sparsity and fusion penalties yields subproblems that each decompose blockwise, with the only inter-block coupling being a one-dimensional rank-allocation step inside the Fantope projection. Joint Graphical Lasso additionally admits a natural proximal-gradient method, since its smooth log-determinant gradient and $\ell_1$-plus-fusion proximal operator both decompose blockwise; running both solvers on Joint Graphical Lasso lets us empirically check the solver-agnostic claim of Corollary 3.4.

Figure 2 reports the ADMM comparison: the CS reduction reduces total path runtime by roughly 75% across all $(K, d)$ settings, with the advantage growing as $d$ increases as expected from the $\mathcal{O}(d^3)$ scaling of the dominant eigendecomposition and Fantope-projection steps. Figure 3 reports the same comparison under a proximal-gradient solver on Joint Graphical Lasso, for which proximal gradient is the natural choice. The matching speedup pattern is the empirical counterpart of Corollary 3.4: the CS reduction is solver-agnostic, and the practical benefit transfers cleanly across first-order methods.

### 5.2 Time to Compute a Single Solution

Figure 2 reports an aggregate speedup averaged over an entire regularization path; this subsection asks where on the path that speedup originates. For each penalty pair $(\lambda_1, \lambda_2)$ we define the computationally sufficient sparsity as the fraction of entries zeroed out by the reduction:

$$s = 1 - \frac{\mathrm{nnz}(R(\mathbb{X}))}{\mathrm{nnz}(\mathbb{X})},$$

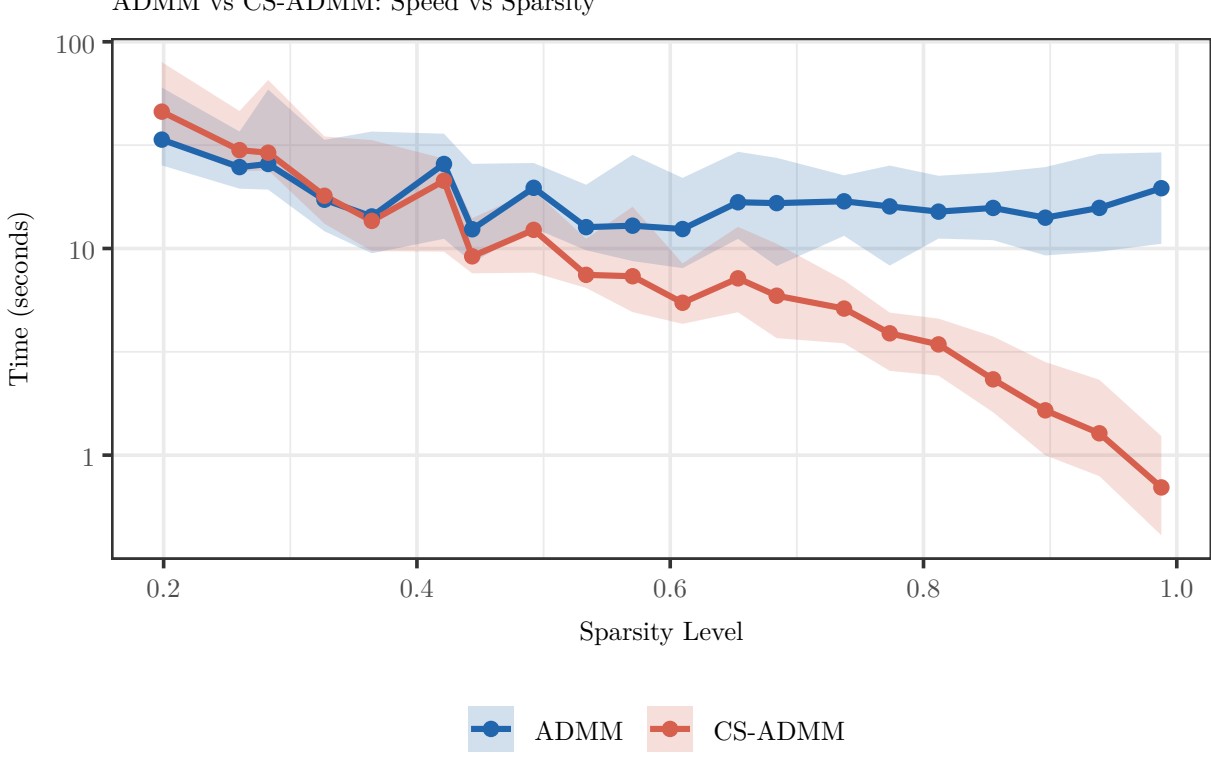

Figure 4: Per-penalty-pair wall-clock time (seconds, log scale) versus CS sparsity $s$ for joint Sparse PCA with Group Lasso ($d = 300$, $K = 3$, 200 penalty pairs per run, 5 runs). Each point is the binned median over 20 equal-width sparsity bins; bands show the 25th–75th percentile range within each bin.

so that $s = 0$ when the reduction does nothing and $s \to 1$ when the problem concentrates into a small active submatrix. We emphasize that $s$ measures the strength of the CS reduction—how much of $\mathbb{X}$ is zeroed before the solver runs—not the sparsity of the resulting estimator.

Larger tuning parameters generally yield larger $s$, but the precise mapping from $(\lambda_1, \lambda_2)$ to $s$ depends on the distribution of entry magnitudes in $\mathbb{X}$: the screening graph $\mathcal{G}_{\mathcal{P}}$ is built by comparing entry magnitudes to thresholds derived from $(\lambda_1, \lambda_2)$, so absolute values of the tuning parameters are only interpretable relative to the scale of the data. We therefore plot runtime against $s$, which is data-adaptive, rather than against $(\lambda_1, \lambda_2)$ directly.

Fixing $d = 300$, $K = 3$, and target rank $r = 2$ for joint Sparse PCA, we record per-pair runtimes over 5 independent runs of 200 penalty configurations each (8 values of $\lambda_1 \times 25$ values of $\lambda_2$). The per-pair speedup grows with $s$: at small $s$ the two solvers are comparable because the cost of constructing the screening graph $\mathcal{G}_{\mathcal{P}}$ is paid regardless of whether any reduction follows; at moderate-to-large $s$, CS-ADMM becomes substantially faster because the dominant $\mathcal{O}(d^3)$ Fantope projection is replaced by cheaper block-wise eigendecompositions on a smaller active block.

## 6  Discussion

In this paper, we present a unified framework for the joint estimation of multiple sparse symmetric matrices, incorporating group and fusion penalties to leverage shared structure across related matrices. By examining a family of methods, including extensions of Graphical Lasso, sparse PCA, and covariance estimation with

$L_p$ and Generalized Lasso penalties, we provide both theoretical insights and practical advancements in computational efficiency.

A key contribution is the identification of a shared computational structure for the joint multiple-matrix problem through the lens of computational sufficiency, which uncovers hidden commonalities among seemingly distinct estimation problems. The projection-based theory we build on is due to Vu (2018b); the multiple-matrix extension and the penalty-specific screening graphs that make it concrete for fusion penalties are what is new here. This unification enhances our theoretical understanding of the optimization challenges and enables the derivation of a universal screening rule that reduces the search space to block diagonal matrices. Our empirical analysis demonstrates that this reduction significantly improves runtime, making the methods more scalable for high-dimensional applications.

**Scope of the invariance condition.** The unified reduction applies to generators $A$ that are diagonal sign conjugation invariant: $A(D\theta D) = A(\theta)$ for every diagonal $\pm 1$ matrix $D$ (Definition 2.1). This symmetry encodes the natural fact that the orientation of any coordinate is arbitrary, so a loss function should not depend on the sign convention chosen for any variable. It is satisfied by all the running examples in this paper—the Graphical Lasso, sparse PCA, sparse covariance, and Ising generators—and more broadly by any generator that depends on $\theta$ only through the absolute values of its off-diagonal entries together with arbitrary functions of its diagonal. The class of such generators is strictly larger than the class of orthogonally invariant generators: generators that depend only on $|\theta_{ij}|$ satisfy our condition but are typically not invariant under general orthogonal conjugation. Moreover, the diagonal sign group is the maximal subgroup of the orthogonal group whose conjugation action on symmetric matrices preserves the support of every $\theta$ entrywise: any strictly larger subgroup contains elements that relabel or mix coordinates, which generically move zero entries to nonzero positions. This maximality is what makes diagonal sign invariance the natural symmetry to exploit when the goal is a sparsity-preserving reduction. Generators or penalties that distinguish positive from negative off-diagonal entries —for instance asymmetric losses, or fusion penalties weighted by signed differences—break this invariance and fall outside our framework.

While these screening rules provide substantial computational benefits, their current limitation lies in the assumption of a common block diagonal structure for all matrices. This may not be optimal in scenarios where matrices exhibit distinct block patterns. A crucial direction for future work is to relax this assumption, allowing for varied block structures across matrices. For instance, in cases with Fused Lasso penalties, where solutions are piecewise constant, we can further refine the feasible set by identifying points of discontinuity within each matrix, thereby enhancing computational efficiency without enforcing uniformity.

In conclusion, our proposed framework offers significant advancements in both theory and practice for sparse matrix estimation. Despite existing limitations, the generality and flexibility of the computational sufficiency approach indicate promising avenues for future extensions. Exploring more adaptable screening rules is likely to yield even greater scalability and applicability in complex high-dimensional data analysis.

### Acknowledgments

We thank the Action Editor and the anonymous reviewers for their careful reading and constructive feedback, which improved the paper. This material is based upon work supported by the National Science Foundation under Grant No. DMS-1916446.

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
