# Computationally Sufficient Reductions for Joint Multiple Matrix Estimators with Sparsity and Fusion: Supplementary Material

**Prateek Sasan**                                                          *sasan.prateek@gmail.com*
*Department of Statistics*
*The Ohio State University*

**Vincent Q. Vu**                                                                *vqv@stat.osu.edu*
*Department of Statistics*
*The Ohio State University*

**Reviewed on OpenReview:** *https://openreview.net/forum?id=KK9RHgSbdp*

## 1 Some Concepts from Computational Sufficiency

To prove the main result, i.e., Theorem 3.3, we will use some concepts introduced by Vu (2018) that we collect here for reference. See Vu (2018) for more details on computational sufficiency.

**Definition 1.1** (Expofam-type Estimator)**.** Let $\mathcal{X}$ be the Euclidean space equipped with the inner product $\langle \cdot, \cdot \rangle$ inducing a norm $\| \cdot \|$. A set-valued estimator $T$ on $\mathcal{X}$ is an *expofam-type* estimator if it has the form

$$T(x) = \arg \min_{\theta} \mathcal{A}(\theta) - \langle x, \theta \rangle + h_C(\theta),$$

where $\mathcal{A}$ is a closed, convex, and proper function called the generator of $T$, and $h_C$ is the support function of the set $C$, i.e.,

$$h_C(\theta) = \max_{z \in C} \langle z, \theta \rangle.$$

**Definition 1.2.** (Vu, 2018) Let $\mathcal{M}$ be a collection of set-valued functions $T : \mathcal{X} \to 2^{\mathcal{T}}$. A function $R : \mathcal{X} \to \mathcal{Y}$ is *computationally sufficient* for $\mathcal{M}$ if for each $T \in \mathcal{M}$, there exists a set-valued function $f_T : \mathcal{Y} \to 2^{\mathcal{T}}$ such that

$$f_T(R(x)) \subseteq T(x) \quad \forall \, x \in \mathcal{X}$$

and $f_T(R(x)) \neq \emptyset$ whenever $T(x) \neq \emptyset$.

To get any meaningful computationally sufficient reduction, the choice of $\mathcal{M}$ is crucial. Using the language of group theory, Vu (2018) used the invariances exhibited by the generators of expofam-type estimators to select the class $\mathcal{M}$. We state some basic definitions from group theory before giving the main results from Vu (2018).

**Definition 1.3** (Group Invariance)**.** Let $\mathcal{G}$ be a compact subgroup of the orthogonal group $\mathcal{O}(\mathcal{X})$. A function $f$ on $\mathcal{X}$ is $\mathcal{G}$-invariant if it is invariant under the action of $\mathcal{G}$ on $\mathcal{X}$, i.e.,

$$f(g \cdot x) = f(x) \text{ for all } x \in \mathcal{X} \text{ and } g \in \mathcal{G}.$$

**Definition 1.4** (Orbitope)**.** The *orbitope* $\text{conv}(\mathcal{G} \cdot x)$ of an element $x$ under $\mathcal{G}$, is defined as the convex hull of the group action, i.e.,

$$\text{conv}(\mathcal{G} \cdot x) = \text{conv}(\{g \cdot x \mid g \in \mathcal{G}\})$$

**Definition 1.5** ($\mathcal{G}$-Majorization)**.** (Eaton & Perlman, 1977) We say $x$ is $\mathcal{G}$-*majorized* by $y$ if $x \in \text{conv}(\mathcal{G} \cdot y)$. This induces a preorder on $\mathcal{X}$ denoted by

$$x \preceq_{\mathcal{G}} y \iff x \in \text{conv}(\mathcal{G} \cdot y).$$

**Theorem 1.6.** *(Vu, 2018, Theorem 2) Let $T$ be an expofam-type estimator with penalty support set $C$, and a generator $A$ that is closed, convex, proper, and $\mathcal{G}$-invariant. If $Q : \mathcal{X} \to \mathcal{X}$ is an orthogonal projection satisfying*

1. *(averaging) $Qx \preceq_{\mathcal{G}} x$ for all $x \in \mathcal{X}$,*

2. *(dual feasibility) $Q(x - C) \subseteq x - C$, and*

3. *(dual invariance) $Q(x - C) \subseteq Qx - C$,*

*then $QT(x) = QT(Qx) \subseteq T(x) \cap T(Qx)$.*

## 2 Proofs of Theorem 3.3 and Corollary 3.4

We will use Theorem 1.6. In the first part of the proof, we will show that the estimators under consideration are expofam-type estimators and identify the group of invariances, its orbitopes, and group majorization. In the second part we derive the form of the penalty support set when the fusion penalty is the Generalized Lasso or Group Lasso penalty. Then in the third part we relate the joint single-linkage thresholding reduction to an orthogonal projection and verify that it satisfies the three numbered conditions of Theorem 1.6.

### 2.1 Part 1: Expofam-type estimators and the induced group majorization

The estimators are of the form

$$
\begin{aligned}
T_A(\mathbb{X}) &= \arg\min_{\Theta} \mathcal{L}_{A,\mathcal{P}}(\mathbb{X}, \Theta, \lambda_1, \lambda_2) \\
&= \arg\min_{\Theta} \left[ \sum_{k=1}^{K} L_A(\mathbb{X}^{(k)}, \Theta^{(k)}, \lambda_1) \right] + \lambda_2 \mathcal{P}(\Theta) \\
&= \arg\min_{\Theta} \left[ \sum_{k=1}^{K} A(\Theta^{(k)}) \right] - \left[ \sum_{k=1}^{K} \langle \mathbb{X}^{(k)}, \Theta^{(k)} \rangle \right] + \left[ \left( \sum_{k=1}^{K} \lambda_1 \|\Theta^{(k)}\|_1 \right) + \lambda_2 \mathcal{P}(\Theta) \right]. \quad (2.1)
\end{aligned}
$$

Now let $\mathcal{X}$ be the $K$-fold Cartesian product of the space of real symmetric $d \times d$ matrices and extend the inner product in the natural way, e.g.

$$
\langle \mathbb{U}, \mathbb{V} \rangle = \sum_{k=1}^{K} \langle \mathbb{U}^{(k)}, \mathbb{V}^{(k)} \rangle .
$$

The second term, from left to right, in (2.1) is an inner product between tuples of matrices. The third term is a sum of penalties and can be expressed as a support function. So we can write (2.1) as

$$
T_A(\mathbb{X}) = \arg\min_{\Theta} \mathcal{A}(\Theta) - \langle \mathbb{X}, \Theta \rangle + h_C(\Theta) ,
$$

which is the form of an expofam-type estimator with generator

$$
\mathcal{A}(\Theta) = \sum_{k=1}^{K} A(\Theta^{(k)})
$$

and penalty support set given by the Minkowski sum

$$
C = \lambda_1 \mathcal{B}_{\infty} \oplus \lambda_2 C_{\mathcal{P}} , \quad (2.2)
$$

where $\mathcal{B}_{\infty}$ is the $L_{\infty}$ ball of radius 1 in $\mathcal{X}$, i.e., tuples of symmetric matrices with entries bounded by 1 in absolute value, and $C_{\mathcal{P}}$ is the penalty support set for the fusion penalty. We will give the specific form of $C_{\mathcal{P}}$ in the next part of the proof.

$A$ is invariant under conjugation by $d \times d$ diagonal sign matrices, so $\mathcal{A}$ inherits a similar invariance:

$$\mathcal{A}(D\Theta^{(1)}D^\top, \ldots, D\Theta^{(K)}D^\top) = \mathcal{A}(\Theta)$$

for all $d \times d$ diagonal sign matrices $D$. The set of all such matrices induces a group of invariances of $\mathcal{A}$ that act by conjugation and we denote it by $\mathcal{G}$. By Lemma 4 of Vu (2018), the orbitopes of $\mathcal{G}$ have the form

$$\mathrm{conv}(\mathcal{G} \cdot \Theta) = \{(W \circ \Theta^{(1)}, \ldots, W \circ \Theta^{(K)}) \mid W \in \mathrm{Cut}_d\},$$

where

$$\mathrm{Cut}_d = \mathrm{conv}\left(\{yy^\top \mid y \in \{-1, +1\}^d\}\right).$$

It follows from the same result that

$$\mathbb{U} \preceq_{\mathcal{G}} \mathbb{V} \iff (\mathbb{U}^{(1)}, \ldots, \mathbb{U}^{(K)}) = (W \circ \mathbb{V}^{(1)}, \ldots, W \circ \mathbb{V}^{(K)}) \tag{2.3}$$

for some $W \in \mathrm{Cut}_d$.

## 2.2  Part 2: Penalty support sets

The penalty support set $C$ is given as the Minkowski sum of an $L_\infty$ ball and the support set of the fusion penalty as in (2.2). We will now derive its form for each of the two cases of fusion penalties.

**Generalized Lasso penalties**  The Generalized Lasso penalty is given by

$$\mathcal{P}_B(\Theta) = \sum_{ij} \|B\Theta_{ij}\|_1 = \sum_{ij} \max_{u: \|u\|_\infty \leq 1} \langle B\Theta_{ij}, u \rangle = \sum_{ij} \max_{u: \|u\|_\infty \leq 1} \langle \Theta_{ij}, B^\top u \rangle.$$

From this expression we can see that

$$C_{\mathcal{P}_B} = \{\mathbb{Z} \mid \mathbb{Z}_{ij} = B^\top u_{ij} \text{ for some } u_{ij} \text{ with } \|u_{ij}\|_\infty \leq 1 \text{ for each } i, j\},$$

and

$$C = \lambda_1 \mathcal{B}_\infty \oplus \lambda_2 C_{\mathcal{P}_B}. \tag{2.4}$$

**Group Lasso penalties**  The Group Lasso penalty is given by

$$\mathcal{P}_{\ell_q}(\Theta) = \sum_{ij} \|\Theta_{ij}\|_q.$$

So the support set is

$$C_{\mathcal{P}_{\ell_q}} = \{\mathbb{Z} \mid \|\mathbb{Z}_{ij}\|_r \leq 1 \text{ for each } i, j\},$$

where $r$ is such that $1/r + 1/q = 1$, and

$$C = \lambda_1 \mathcal{B}_\infty \oplus \lambda_2 C_{\mathcal{P}_{\ell_q}}. \tag{2.5}$$

Note that $C$ in both (2.4) and (2.5) has *separable structure* in the sense that

$$\{\mathbb{Z}_{ij} \mid \mathbb{Z} \in C\}$$

does not depend on $(i, j)$. We will exploit this fact in the next part of the proof.

### 2.3 Part 3: Joint single-linkage thresholding reduction

The joint single-linkage thresholding reduction is indeed an orthogonal projection on $\mathcal{X}$. Fix $(\mathbb{X}, \lambda_1, \lambda_2)$ and let

$$W = \mathrm{SLC}_{\mathcal{P}}(\mathbb{X}, \lambda_1, \lambda_2).$$

Then

$$\mathrm{SLT}_{\mathcal{P}}(\mathbb{X}, \lambda_1, \lambda_2)(\mathbb{Z}) = (W \circ \mathbb{Z}^{(1)}, \ldots, W \circ \mathbb{Z}^{(K)}).$$

This is a linear transformation in $\mathbb{Z}$. It is self-adjoint, because the Hadamard product is self-adjoint, and it is idempotent, because $W \circ W = W$ for any binary matrix $W$. Now we verify the three numbered conditions of Theorem 1.6. Conditions 1 and 3 are handled in the same way for both the Generalized Lasso and Group Lasso penalties, so we check those first. Verifying condition 2 is more involved, so we will address it separately for each penalty.

**Condition 1 (Averaging)**  By (2.3), we need to check that $W \in \mathrm{Cut}_d$. This follows from Lemma 5 of Vu (2018), because $\mathrm{SLC}_{\mathcal{P}}$ produces a binary clustering matrix and any such matrix must satisfy the ultrametric inequality.

**Condition 3 (Dual Invariance)**  By Lemma 3 of Vu (2018), it suffices to check that if $\mathbb{Z} \in C$, then

$$(W \circ \mathbb{Z}^{(1)}, \ldots, W \circ \mathbb{Z}^{(K)}) \in C.$$

This follows immediately from (2.4) and (2.5) for the Generalized Lasso and Group Lasso penalties respectively.

#### 2.3.1 Condition 2 (Dual Feasibility)

We need to verify that for each $\mathbb{Z} \in C$, there exists $\mathbb{U} \in \mathcal{B}_{\infty}$ and $\mathbb{V} \in C_{\mathcal{P}}$ such that

$$\left(W \circ (\mathbb{X}^{(1)} - \mathbb{Z}^{(1)}), \ldots, W \circ (\mathbb{X}^{(K)} - \mathbb{Z}^{(K)})\right) = \left(\mathbb{X}^{(1)} - \lambda_1 \mathbb{U}^{(1)} - \lambda_2 \mathbb{V}^{(1)}, \ldots, \mathbb{X}^{(K)} - \lambda_1 \mathbb{U}^{(K)} - \lambda_2 \mathbb{V}^{(K)}\right).$$
$$(2.6)$$

Since $C$ has separable structure, we can consider each pair $(i, j)$ separately in verifying the condition (2.6). If $W_{ij} = 1$, then the condition is trivially satisfied. So we need to verify it for pairs $(i, j)$ such that $W_{ij} = 0$. Note that $W_{ij} = 0$ implies that our penalty specific graph does not have an edge between the pair $(i, j)$. So we will show that if there is no edge between the pair $(i, j)$, then

$$\mathbb{X}_{ij} = \lambda_1 \mathbb{U}_{ij} + \lambda_2 \mathbb{V}_{ij} \tag{2.7}$$

for some $\mathbb{U} \in \mathcal{B}_{\infty}$ and $\mathbb{V} \in C_{\mathcal{P}}$. We will verify this for the Group Lasso and Generalized Lasso penalties separately.

**Group Lasso penalties**  Our construction of the penalty specific graph for the Group Lasso penalty means that

$$W_{ij} = 0 \implies \sum_{k=1}^{K} \max(|\mathbb{X}_{i,j}^{(k)}| - \lambda_1, 0)^r \leq \lambda_2^r. \tag{2.8}$$

If this is the case, then (2.7) is satisfied with

$$\mathbb{U}_{ij}^{(k)} = \begin{cases} \mathbb{X}_{ij}^{(k)}/\lambda_1 & \text{if } |\mathbb{X}_{ij}^{(k)}| \leq \lambda_1, \\ 1 & \text{if } \mathbb{X}_{ij}^{(k)} > \lambda_1, \\ -1 & \text{if } \mathbb{X}_{ij}^{(k)} < -\lambda_1, \end{cases}$$

and

$$\mathbb{V}_{ij}^{(k)} = \begin{cases} 0 & \text{if } |\mathbb{X}_{ij}^{(k)}| \leq \lambda_1, \\ (\mathbb{X}_{ij}^{(k)} - \lambda_1)/\lambda_2 & \text{if } \mathbb{X}_{ij}^{(k)} > \lambda_1, \\ (\mathbb{X}_{ij}^{(k)} + \lambda_1)/\lambda_2 & \text{if } \mathbb{X}_{ij}^{(k)} < -\lambda_1. \end{cases}$$

This verifies that (2.8) ensures that the dual feasibility condition is satisfied.

**Generalized Lasso penalties** This part of the proof was inspired by Yang et al. (2015). For the Generalized Lasso penalties, (2.7) can be thought of as solving a linear feasibility problem. Let

$$H = \{s \in \mathbb{R}^K \mid s = \lambda_1 z_1 + \lambda_2 B^\top z_2, \|z_1\|_\infty \leq 1, \|z_2\|_\infty \leq 1\}.$$

We need to show that

$$W_{ij} = 0 \implies \mathbb{X}_{ij} \in H.$$

Now $H$ is a closed convex set and its support function is given by

$$f(y) = \max_{\|z_1\|_\infty \leq 1, \|z_2\|_\infty \leq 1} \langle y, \lambda_1 z_1 + \lambda_2 B^\top z_2 \rangle = \lambda_1 \|y\|_1 + \lambda_2 \|By\|_1.$$

By the Fenchel–Moreau Theorem, the convex conjugate of this support function gives us a variational characterization of the convex indicator of $H$:

$$f^*(x) = \max_y \langle x, y \rangle - \lambda_1 \|y\|_1 - \lambda_2 \|By\|_1 = \begin{cases} 0 & \text{if } x \in H, \\ +\infty & \text{otherwise.} \end{cases}$$

Since

$$y \mapsto \langle x, y \rangle - \lambda_1 \|y\|_1 - \lambda_2 \|By\|_1$$

is a positive homogeneous function, we see that

$$\mathbb{X}_{ij} \in H \iff \langle \mathbb{X}_{ij}, y \rangle - \lambda_1 \|y\|_1 - \lambda_2 \|By\|_1 \leq 0 \text{ for all } y \in \mathbb{R}^K.$$

This is exactly the edge-exclusion condition $g_{ij}(y) \leq 0$ for the penalty-specific graph of the Generalized Lasso penalty (Section 3.3 of the main paper). So if $W_{ij} = 0$, then the edge is excluded and $\mathbb{X}_{ij} \in H$.

### 2.4 Proof of Corollary 3.4

By Equation (3.1), the original problem has a minimizer of the form $\mathrm{SLT}_{\mathcal{P}}(\mathbb{X}, \lambda_1, \lambda_2)(\widehat{\Theta})$ for some $\widehat{\Theta}$ minimizing the reduced objective on $R(\mathbb{X})$, and the image of $\mathrm{SLT}_{\mathcal{P}}(\mathbb{X}, \lambda_1, \lambda_2)$ is exactly $\mathcal{B}$, so (3.2) contains such a minimizer. Since $\mathcal{B}$ is a linear subspace, the (sub)gradient of the restriction of a convex function to $\mathcal{B}$ is the orthogonal projection of the original (sub)gradient onto $\mathcal{B}$, hence lies in $\mathcal{B}$. The proximal operator of the restriction is by definition a minimizer over $\mathcal{B}$ and thus also lies in $\mathcal{B}$. Any first-order method on (3.2) combines these operations with linear combinations, which preserve $\mathcal{B}$.

## 3 Algorithms

In this section we present algorithms for computing the joint estimators of the form

$$T_A(\mathbb{X}) = \arg\min_\Theta \left[ \sum_{k=1}^K A(\Theta^{(k)}) - \langle \mathbb{X}^{(k)}, \Theta^{(k)} \rangle + \lambda_1 \|\Theta^{(k)}\|_1 \right] + \lambda_2 \mathcal{P}(\Theta).$$

We first describe a general alternating direction method of multipliers (ADMM) (Boyd et al., 2011, Chapter 3, Section 3.1.1) framework for the family and specialize it to Joint Sparse PCA and Joint Graphical Lasso with a Group Lasso fusion penalty. We then describe an alternative proximal gradient method specialized to Joint Graphical Lasso with the same fusion penalty.

### 3.1 ADMM Algorithm

#### 3.1.1 General Framework

The ADMM algorithm decouples the objective function defining the estimator into two parts by defining a new equality constraint:

$$\begin{aligned} \underset{\Theta, \beta}{\text{minimize}} \quad & \left[ \sum_{k=1}^K A(\Theta^{(k)}) - \langle \mathbb{X}^{(k)}, \Theta^{(k)} \rangle + \lambda_1 \|\beta^{(k)}\|_1 \right] + \lambda_2 \mathcal{P}(\beta) \\ \text{subject to} \quad & \Theta^{(k)} - \beta^{(k)} = 0, k = 1, \ldots, K \end{aligned}$$

This form of the problem is equivalent to the original problem, but the ADMM algorithm utilizes an augmented Lagrangian for the equality constraint:

$$L_\rho(\Theta, \beta, U) = \left[\sum_{k=1}^K A(\Theta^{(k)}) - \langle \mathbb{X}^{(k)}, \Theta^{(k)}\rangle + \lambda_1\|\beta^{(k)}\|_1\right] + \lambda_2 \mathcal{P}(\beta)$$
$$+ \frac{\rho}{2}\sum_{k=1}^K\left(\|\Theta^{(k)} - \beta^{(k)} + U^{(k)}\|_F^2 - \|U^{(k)}\|_F^2\right).$$

The ADMM algorithm cycles through updates for the two primal variables followed by an update of a dual variable. The $(t+1)$th iteration of the algorithm involves the following three steps:

$$\Theta_{t+1}^{(k)} = \arg\min_{\Theta^{(k)}}\left(A(\Theta^{(k)}) - \langle \mathbb{X}^{(k)}, \Theta^{(k)}\rangle + (\rho/2)\|\Theta^{(k)} - \beta_t^{(k)} + U_t^{(k)}\|_F^2\right) \quad k = 1, \ldots, K$$

$$\beta_{t+1} = \arg\min_\beta\left(\lambda_1 \sum_{k=1}^K\|\beta^{(k)}\|_1 + \lambda_2 \mathcal{P}(\beta) + (\rho/2)\sum_{k=1}^K\|\Theta_{t+1}^{(k)} - \beta^{(k)} + U_t^{(k)}\|_F^2\right)$$

$$U_{t+1}^{(k)} = U_t^{(k)} + \Theta_{t+1}^{(k)} - \beta_{t+1}^{(k)} \quad k = 1, \ldots, K.$$

The first two steps involve computing proximal operators associated with the generator and penalty functions, respectively (Parikh & Boyd, 2014). Writing

$$\text{prox}_f(x) = \arg\min_y \frac{1}{2}\|x - y\|_F^2 + f(y)$$

for the proximal operator, the ADMM updates can be written as

$$\Theta_{t+1}^{(k)} = \text{prox}_{A/\rho}\left(\frac{\mathbb{X}^{(k)}}{\rho} + \beta_t^{(k)} - U_t^{(k)}\right) \quad k = 1, \ldots, K$$

$$\beta_{t+1} = \text{prox}_{\lambda_1/\rho\|\cdot\|_1 + \lambda_2/\rho\mathcal{P}}(\Theta_{t+1} + U_t)$$

$$U_{t+1}^{(k)} = U_t^{(k)} + \Theta_{t+1}^{(k)} - \beta_{t+1}^{(k)} \quad k = 1, \ldots, K.$$

Note that the update for $\Theta$ is separable across the $K$ matrices in the tuple, so it can be computed in parallel. The update for $\beta$ involves the proximal operator of the sum of the $L_1$ and fusion penalties and this is where *strength is borrowed* across the $K$ related matrices.

Algorithm 1 shows the general algorithm.

### 3.1.2 Specialization to Joint Sparse PCA and Joint Graphical Lasso with Group Fusion

Here, we provide details of the algorithm for the Joint Sparse PCA and Joint Graphical Lasso with a Group Lasso fusion penalty. To specialize the general algorithm, we need to derive the proximal operators of the generators and fusion penalties associated with these estimators.

**Proximal Operator for the Sparse PCA Generator.** The generator for Sparse PCA is the convex indicator of the Fantope i.e.,

$$A(\Theta^{(k)}) = \begin{cases} 0 & \text{if } \Theta^{(k)} \in \mathcal{F}^r \\ +\infty & \text{otherwise.} \end{cases}$$

Thus, the proximal operator is the projection onto the Fantope i.e.,

$$\text{prox}_{A/\rho}\left(\frac{\mathbb{X}^{(k)}}{\rho} + \beta_t^{(k)} - U_t^{(k)}\right) = \mathcal{P}_{\mathcal{F}^r}\left(\frac{\mathbb{X}^{(k)}}{\rho} + \beta_t^{(k)} - U_t^{(k)}\right)$$

This projection, given an input $Z$, can be found as follows:

---

**Algorithm 1** General ADMM Algorithm

---

**Require:** (Input) $\mathbb{X}^{(1)}, \ldots, \mathbb{X}^{(K)}$, (Generator) $A$, (Fusion Penalty Type) $\mathcal{P}$, (Fusion Penalty Parameter) $\lambda_2$, ($L_1$ Penalty) $\lambda_1$, (Step Size) $\rho > 0$, (Tolerance) $\epsilon$

1: Set $\beta_0^{(k)} = 0$, $\beta_{-1}^{(k)} = 0$, $U_0^{(k)} = 0$, for $k = 1, \ldots, K$
2: Set $\Theta_0^{(k)} = I$, for $k = 1, \ldots, K$
3: Set $t = 0$
4: **while** $\max \left( \sum_{k=1}^{K} \|\Theta_t^{(k)} - \beta_t^{(k)}\|_F^2, \rho^2 \sum_{k=1}^{K} \|\beta_t^{(k)} - \beta_{t-1}^{(k)}\|_F^2 \right) \geq \epsilon$ **do**
5:      $\Theta_{t+1}^{(k)} = \text{prox}_{A/\rho} \left( \frac{\mathbb{X}^{(k)}}{\rho} + \beta_t^{(k)} - U_t^{(k)} \right)$    $k = 1, \ldots, K$
6:      $\beta_{t+1}^{(k)} = \text{prox}_{\lambda_1/\rho \|\cdot\|_1 + \lambda_2/\rho \mathcal{P}} \left( \Theta_{t+1}^{(k)} + U_t^{(k)} \right)$
7:      $U_{t+1}^{(k)} = U_t^{(k)} + \Theta_{t+1}^{(k)} - \beta_{t+1}^{(k)}$    $k = 1, \ldots, K$
8:      Set $t = t + 1$
9: **end while**
10: Return $\Theta_t$

---

1. Let $Z = \sum_i \gamma_i z_i z_i^\top$ be the eigendecomposition of $Z$.

2. Then $\mathcal{P}_{\mathcal{F}^r}(Z) = \sum_i \gamma_i^+(\theta) z_i z_i^\top$ where $\gamma_i^+(\theta) = \min(\max(\gamma_i - \theta, 0), 1)$ and $\sum_i \gamma_i^+(\theta) = r$.

See Vu et al. (2013, Lemma 1) for more details.

**Proximal Operator for the Graphical Lasso Generator.** In the family form of (3), the generator for the Graphical Lasso is

$$
A(\theta) = \begin{cases} -\log \det(-\theta) & \text{if } -\theta \succeq 0 \\ +\infty & \text{otherwise,} \end{cases}
$$

so that $\theta$ is the *negative* of the precision matrix (see (2.6) in the main paper). Its proximal operator can be computed by an eigenvalue transformation: given an input $Z$ with eigendecomposition $Z = \sum_i \gamma_i z_i z_i^\top$,

$$
\text{prox}_{A/\rho}(Z) = \sum_i \xi_i^\star z_i z_i^\top \quad \text{where} \quad \xi_i^\star = \frac{\gamma_i - \sqrt{\gamma_i^2 + 4/\rho}}{2}.
$$

The eigenvalues $\xi_i^\star$ are non-positive, so $\text{prox}_{A/\rho}(Z) \preceq 0$ as required. See Danaher et al. (2014); Witten & Tibshirani (2009) for more details.

**Proximal Operator for $L_1$ Lasso + $L_2$ Group Lasso Fusion Penalty.** The sum of the $L_1$ and $L_2$ Group Lasso fusion penalties is

$$
f(\Theta) = \lambda_1 \sum_{k=1}^{K} \|\Theta^{(k)}\|_{1,1} + \lambda_2 \mathcal{P}_{\ell_2}(\Theta)
$$

$$
= \lambda_1 \sum_{k=1}^{K} \|\Theta^{(k)}\|_{1,1} + \lambda_2 \sum_{ij} \|\Theta_{ij}\|_2.
$$

The corresponding proximal operator in the ADMM algorithm is given by

$$
\text{prox}_{\lambda_1/\rho \|\cdot\|_1 + \lambda_2/\rho \mathcal{P}}(\mathbb{Z})_{ij}^{(k)} = \mathcal{S}_{\lambda_1/\rho}(\mathbb{Z}_{ij}^{(k)}) \max \left( 1 - \frac{\lambda_2}{\rho \sqrt{\sum_{k=1}^{K} \mathcal{S}_{\lambda_1/\rho}(\mathbb{Z}_{ij}^{(k)})^2}}, 0 \right).
$$

Here, $\mathcal{S}_{\lambda_1/\rho}(\cdot)$ is the soft-thresholding operator (Parikh & Boyd, 2014, Chapter 6, Section 6.5.2). See Danaher et al. (2014) for more details.

### 3.2 Proximal Gradient Algorithm for Joint Graphical Lasso

#### 3.2.1 General Framework

We consider the joint estimator for Graphical Lasso with Group Lasso fusion penalty. We work directly with the precision-matrix form of the Joint Graphical Lasso loss, rather than the family form used elsewhere in the paper; the two are equivalent up to a sign flip of the data input, but the precision-matrix form is more convenient for stating the gradient and PSD projection steps below. The optimization problem is

$$\arg\min_{\Theta \succeq 0} \left[ \sum_{k=1}^{K} A\left(\Theta^{(k)}\right) + \langle \mathbb{X}^{(k)}, \Theta^{(k)} \rangle + \lambda_1 \|\Theta^{(k)}\|_{1,1} \right] + \lambda_2 \mathcal{P}_{\ell_2}(\Theta) \tag{3.1}$$

where the generator $A(\Theta^{(k)}) = -\log\det(\Theta^{(k)})$ and $\mathcal{P}_{\ell_2}(\Theta) = \sum_{ij} \|\Theta_{ij}\|_2$ is the Group Lasso fusion penalty.

We solve (3.1) via a proximal gradient method (Parikh & Boyd, 2014), splitting the objective into a smooth part

$$f(\Theta) = \sum_{k=1}^{K} \left[ -\log\det\left(\Theta^{(k)}\right) + \langle \mathbb{X}^{(k)}, \Theta^{(k)} \rangle \right]$$

and a non-smooth part $g(\Theta) = \lambda_1 \sum_{k=1}^{K} \|\Theta^{(k)}\|_{1,1} + \lambda_2 \mathcal{P}_{\ell_2}(\Theta)$. Each iteration of the algorithm consists of four steps.

**Step 1: Gradient-descent step on $f$.** The gradient of $f$ with respect to $\Theta^{(k)}$ is $\nabla_k f = \mathbb{X}^{(k)} - (\Theta^{(k)})^{-1}$, giving the update

$$\Theta^{(k)} \leftarrow \Theta^{(k)} + \alpha\left(\Theta^{(k)}\right)^{-1} - \alpha\mathbb{X}^{(k)} \quad k = 1, \dots, K$$

with step size $\alpha > 0$. The result is then projected onto the positive semidefinite cone by clipping any negative eigenvalues to a small positive constant $\varepsilon > 0$.

**Step 2: Proximal step for the $\ell_1$ penalty.** Element-wise soft-thresholding with threshold $\alpha\lambda_1$:

$$\Theta_{ij}^{(k)} \leftarrow \mathcal{S}_{\alpha\lambda_1}\left(\Theta_{ij}^{(k)}\right) \quad k = 1, \dots, K$$

where $\mathcal{S}_\tau(x) = \text{sign}(x)\max(|x| - \tau, 0)$ is the soft-thresholding operator (Parikh & Boyd, 2014).

**Step 3: Proximal step for the Group Lasso fusion penalty.** Group soft-thresholding applied jointly across all $K$ matrices at each entry $(i, j)$:

$$\Theta_{ij}^{(k)} \leftarrow \Theta_{ij}^{(k)} \cdot \max\left(1 - \frac{\alpha\lambda_2}{\|\Theta_{ij}\|_2}, 0\right) \quad k = 1, \dots, K$$

where $\|\Theta_{ij}\|_2 = \sqrt{\sum_{k=1}^{K}(\Theta_{ij}^{(k)})^2}$ is the $\ell_2$ norm of the $K$-vector of $(i, j)$ entries across matrices.

**Step 4: PSD re-projection.** Steps 2 and 3 may violate positive semidefiniteness. We re-project each $\Theta^{(k)}$ onto the PSD cone by computing its eigendecomposition and clipping negative eigenvalues to $\varepsilon$.

The algorithm iterates until the change in iterates falls below a tolerance $\delta$ or a maximum number of iterations is reached. Algorithm 2 summarizes the procedure.

#### 3.2.2 Integration with the CS Reduction

The CS reduction applies before the proximal gradient loop. Following the general recipe used in the main paper for Joint Sparse PCA, we compute $R(\mathbb{X})$ and restrict the feasible set to tuples $\Theta \in \mathcal{B}_{\mathcal{P}}(\mathbb{X}, \lambda_1, \lambda_2)$, whose matrices are block-diagonal with blocks given by the connected components $C_1, \dots, C_m$ of the penalty graph $\mathcal{G}_{\mathcal{P}}(\mathbb{X}, \lambda_1, \lambda_2)$.

Within the proximal gradient loop, each of the four steps above is applied block-wise:

---

**Algorithm 2** Proximal Gradient for Joint Graphical Lasso with Group Lasso

---

**Require:** (Input) $\mathbb{X}^{(1)}, \ldots, \mathbb{X}^{(K)}$, ($L_1$ Penalty) $\lambda_1$, (Fusion Penalty) $\lambda_2$, (Step Size) $\alpha > 0$, (PSD Floor) $\varepsilon > 0$, (Tolerance) $\delta$

1: Set $\Theta_0^{(k)} = I$, for $k = 1, \ldots, K$
2: Set $t = 0$
3: **while** $\sum_{k=1}^{K} \|\Theta_t^{(k)} - \Theta_{t-1}^{(k)}\|_F \geq \delta$ **do**
4:     **for** $k = 1, \ldots, K$ **do**           $\triangleright$ Gradient step on $f = \sum_k [-\log\det(\Theta^{(k)}) + \langle \mathbb{X}^{(k)}, \Theta^{(k)} \rangle]$
5:         $V^{(k)} \leftarrow \Theta_t^{(k)} + \alpha \left(\Theta_t^{(k)}\right)^{-1} - \alpha \mathbb{X}^{(k)}$
6:         $V^{(k)} \leftarrow \mathcal{P}_\varepsilon\left(V^{(k)}\right)$         $\triangleright$ PSD projection: clip eigenvalues to $[\varepsilon, \infty)$
7:     **end for**
8:     **for** $k = 1, \ldots, K$ **do**         $\triangleright$ Proximal step: $\ell_1$ soft threshold
9:         $V^{(k)} \leftarrow \mathcal{S}_{\alpha\lambda_1}\left(V^{(k)}\right)$
10:     **end for**
11:     $T \leftarrow \sqrt{\sum_{k=1}^{K} V^{(k)} \odot V^{(k)}}$         $\triangleright$ Entry-wise $\ell_2$ norms across matrices
12:     **for** $k = 1, \ldots, K$ **do**         $\triangleright$ Proximal step: Group Lasso fusion threshold
13:         $\Theta_{t+1}^{(k)} \leftarrow V^{(k)} \odot \max\left(\mathbf{0}, \mathbf{1} - \frac{\alpha\lambda_2}{T}\right)$
14:     **end for**
15:     **for** $k = 1, \ldots, K$ **do**         $\triangleright$ PSD re-projection after thresholding
16:         $\Theta_{t+1}^{(k)} \leftarrow \mathcal{P}_\varepsilon\left(\Theta_{t+1}^{(k)}\right)$
17:     **end for**
18:     $t \leftarrow t + 1$
19: **end while**
20: **return** $\Theta_t$

---

- **Step 1**: the matrix inverse and eigendecomposition in the gradient-descent step are computed independently for each block of size $|C_b|$, reducing the dominant cost from $\mathcal{O}(Kd^3)$ per iteration to $\mathcal{O}(K\sum_b |C_b|^3)$.

- **Steps 2–3**: soft-thresholding and group thresholding are applied only to entries $(i,j)$ with $i, j \in C_b$ for some active block $b$; all other entries are identically zero by construction.

- **Step 4**: PSD re-projection is performed block-wise, again at cost $\mathcal{O}(K\sum_b |C_b|^3)$.

When $\lambda_2$ is large enough to decompose the problem into small blocks, the savings are substantial: if all $m$ components have equal size $d/m$, the per-iteration cost drops by a factor of $m^2$ relative to the full problem.

## 4 Additional Details for the Experiments

### 4.1 ADMM Simulation

We provide a comprehensive explanation of the simulation procedure used in our experiments. The goal of the simulations is to evaluate the computational performance with and without computationally sufficient reductions.

**Code**   The implementation is available at `https://github.com/prateeksasan1/cs-jmme`

We detail the data generation process, the estimation procedures, and the experimental setup, including the specific parameters used in the simulations.

**Compute Environment**   All experiments were performed on a compute cluster with 1 node, 32 cores, and 64 GB memory.

**Generating Base Covariance Matrix**  The code begins by simulating a base covariance matrix, which serves as a starting point for class-specific covariance matrices. In this setup, a matrix of size $p \times r$ is generated, where $p$ is the total number of features and $r$ is a parameter controlling the rank of the base matrix (set to 5 in this experiment). From the total $p$ dimensions, a subset of indices of size sp $= 0.2 \times p$ is randomly selected (i.e., 20% of the features). For these selected positions, values are drawn from a standard normal distribution to populate the matrix. This matrix is then orthogonalized using the Gram-Schmidt process to ensure linear independence among columns. The base covariance matrix is computed by multiplying the orthogonalized matrix with its transpose, resulting in a symmetric positive semi-definite matrix.

**Adding Class-Specific Perturbations**  Next, class-specific covariance matrices are generated by adding structured perturbations to this base covariance matrix. For each class, two sets of fp $= 2$ features are selected randomly to introduce the perturbations. Two vectors, each of length $p$, are generated, where only the selected positions are filled with normally distributed values. These vectors are then used to construct a perturbation matrix via an outer product. To ensure the matrix remains positive semi-definite, eigenvalue decomposition is applied to the perturbation matrix, and any negative eigenvalues are shifted to non-negative values. The final class-specific covariance matrix is constructed by adding this adjusted perturbation matrix and an identity matrix to the base covariance matrix. The identity matrix helps stabilize the covariance matrix by ensuring a stronger diagonal structure. This process is repeated for each class, resulting in unique covariance matrices for all classes (with the number of classes set to 2, 3, or 5, depending on the simulation setup).

**Simulating Sample Covariance Matrices**  After constructing the class-specific covariance matrices, synthetic data is generated using these matrices. A sample size $n = 500$ is used to generate multivariate normal data for each class. The sample covariance matrices are computed from this synthetic data, which will serve as input for the subsequent optimization steps.

**Parameter Tuning for ADMM and CS-ADMM**  The sample covariance matrices are analyzed to determine appropriate penalty parameters ($\lambda$ values) for the optimization. A sorted sequence of non-zero off-diagonal elements from the sample covariance matrices is used to generate $\lambda$ values for ADMM. Additionally, for each $\lambda$, a set of knot points is determined, which helps refine the optimization process. These knots are used to select the second penalty parameter, forming a grid of parameter values for tuning.

**Optimization Process**  Both ADMM and CS-ADMM algorithms are run on the sample covariance matrices using the grid of $\lambda$ values. The algorithms are set to perform up to 250 iterations with a stopping criterion based on a tolerance level.

**Simulation and Result Collection**  The experiment is conducted across a variety of dimensions $(100, 200, 300, 400, 500)$ and class configurations $(2, 3, 5$ classes$)$. For each combination of dimensions and class numbers, the process is repeated, and the total runtime for both ADMM and CS-ADMM is recorded. The results are saved into a data frame, which is exported for further analysis.

## 4.2 Proximal Gradient Simulation

We use the same data-generating process as the ADMM simulation in Section 4.1: the base covariance is constructed from a sparse rank-5 factor model, and each class-specific covariance is obtained by adding a low-rank, positive-semidefinite perturbation. We draw $n = 500$ observations from each class and compute sample covariance matrices $\mathbb{X}^{(k)}$. We consider $K \in \{2, 3\}$ and $d \in \{100, 150, 200, 250, 300, 400, 500\}$, with 20 independent replicates per setting. Each replicate computes the full regularization path over $5 \times 20 = 100$ penalty configurations. We run both the standard proximal gradient algorithm and its CS-accelerated variant with `max_steps` $= 50$ and tolerance $\delta = 0.001$, with no warm start between $\lambda_2$ values so that both methods receive identical initializations.

Figure 3 reports the total wall-clock time per replicate. Lines show the median over 20 runs; shaded bands show the 25th–75th percentile range. Across all settings the CS reduction yields substantial speedups, with the advantage growing with $d$ as expected from the $\mathcal{O}(d^3)$ scaling of the gradient and projection steps.