# OpenReview forum: "Computationally Sufficient Reductions for Joint Multiple Matrix Estimators with Sparsity and Fusion"
_TMLR — Accepted by TMLR_

### Review · Reviewer_mE5f · 2026-02-26

**Summary Of Contributions:**

The authors address the problem of joint multiple matrix estimation. In this case, given $K$ samples $\mathbb{X}^{(1)}, \ldots, \mathbb{X}^{(K)}$ from the same statistical model, but for potential different parameters $\Theta_0^{(1)}, \ldots, \Theta_0^{(K)}$, one can estimate the parameters by finding

$$
\min_{\Theta^{(1)},\ldots, \Theta^{(K)}}\quad \sum_{k=1}^K L_A(\mathbb{X}^{(k)}, \Theta^{(k)}, \lambda_1)
$$

where $\lambda_1$ represents a hyperparameter for the statistical model. The above problem does not exploit any known relations between $\Theta^{(1)},\ldots, \Theta^{(K)}$ and decouples; in fact, it can be solved separately for each $\Theta^{(k)}$. The authors consider then a *fusion penalty*

$$
	\mathcal{P} = \mathcal{P}(\Theta^{(1)}, \ldots, \Theta^{(K)})
$$

that encodes known structural constraints across the parameters, to then solve

$$
	\min_{\Theta^{(1)},\ldots, \Theta^{(K)}}\quad \sum_{k=1}^K L_A(\mathbb{X}^{(k)}, \Theta^{(k)}, \lambda_1) + \lambda_2 \\mathcal{P}(\Theta^{(1)}, \ldots, \Theta^{(K)}).
$$

where $\lambda_2$ is a hyperparameter. The above problem is no longer separable, and the computational challenges for solving it could be substantial.

To mitigate this effect, the authors propose to leverage the theory of *statistical sufficiency*. By analogy with the concept of *statistical sufficiency*, this implies that the solution depends on the data only through a function of it. The authors prove that for a broad class of (negative log-)likelihoods $L_A$, c.f. Sections 2.2 and 2.3, and specific choices of fusion penalty $\mathcal{P}$, i.e., the graph LASSO and the generalized LASSO, a Joint Single-Linkage Thresholding is computationally sufficient. Furthermore, Theorem 3.3 shows that the optimal solution obtained for the thresholded data must have the same sparsity pattern as the thresholded data; by optimizing only over the non-zero entries, there can be a substantial computational gain and improvements in covergence, which the authors exemplify in numerical experiments.

**Additional Comments:**

No comments.

**Audience:**

Yes

**Audience Explanation:**

The problem of matrix estimation is a classical problem where one typically attempts to design a regularizer promoting a suitable structure. In the problem of joint matrix estimation, which is the focus of the manuscript, the regularizer now must promote structure across the estimated matrices. Similarly, the concept of computational sufficiency is of interest for the community, and the manuscript shows a clear use of these ideas to improve computational efficiency in joint matrix estimation problems.

**Broader Impact Concerns:**

There are no broader impact concerns.

**Claims And Evidence:**

Yes

**Claims Explanation:**

The main contributions of the paper are theoretical. The authors provide a rigorous proof of the main theorem of the manuscript, Theorem 3.3, in Section 2 of the Supplementary Material. The numerical experiments show that their approach yields substantial improvements over a straightforward ADMM implementation to solve the problem.

**Requested Changes:**

I have a few comments.

1. Definition 3.2. should be reworded. At first, it is not clear what is being defined. As a suggestion, you could start by saying that "A joint single-linkage thresholding is...".
2. While Definition 3.2 is in Section 3.2, it references a construction in Section 3.3. Perhaps you could define a joint single-linkage thresholding as a function of a graph, to then state that there exists a graph such that it is computationally sufficient?
3. Figure 1 reflects average times for different choices of penalty parameters. Therefore, the curves should have error bars.
4. In the experiments, the implicit effect of the thresholding parameters is hidden. For some choices of $\lambda_1$ and $\lambda_2$ the thresholded data will be very sparse, and thus the computational gains should be much larger in this case. As the sparsity decreases, the performance of the method may become closer to that of ADMM. For this reason, I suggest making plots of execution time vs sparsity of the thresholded data.

---

> ### Author Response · Authors · 2026-04-13
>
> **Definition 3.2 wording and organization**
>
> We agree. We will reword to lead with "A joint single-linkage thresholding is..." as suggested, and define it as a function of a generic graph $\mathcal{G}$. Section 3.3 will then provide the penalty-specific graph constructions. This removes the forward reference and separates the general mechanism from the specific constructions.
>
> **Error bars in Figure 1**
>
> As noted in General Comment #2, the caption is unclear: each point reports average total runtime over a full regularization path, not a single solve. We will revise the caption and add error bars to show variability across runs.
>
> **Execution time vs. sparsity plots**
>
> We will add plots showing execution time as a function of the sparsity of the thresholded data, directly illustrating when the reduction provides the most benefit. This also addresses Reviewer SJLM's question about when the method helps.

---

> ### Author Response · Authors · 2026-05-05
> **Changes in response to review**
>
> 1. **Definition 3.2 wording and structure.** Reworded as suggested,
>    leading with "A joint single-linkage thresholding is..." and
>    defined as a function of a generic graph $\mathcal{G}$.
> 2. **Figure error bars.** The full-path runtime figure (now Figure 2)
>    shows the median runtime over 20 independent runs with the
>    25th--75th percentile range as a shaded band. The caption was also
>    revised to make explicit that each point reports the total
>    wall-clock time over a $5 \times 20$ regularization path of 100
>    penalty configurations.
> 3. **Execution time vs. sparsity of the thresholded data.** Added
>    as Figure 4 in the new Sec. 5.2 ("Time to Compute a Single
>    Solution"). For each penalty pair $(\lambda_1, \lambda_2)$ we plot
>    per-pair wall-clock time against the *computationally sufficient
>    sparsity* $s = 1 - \mathrm{nnz}(R(X)) / \mathrm{nnz}(X)$ — the
>    fraction of input entries zeroed by the reduction (not the
>    sparsity of the resulting estimator). Runtime savings grow
>    monotonically with $s$: CS-ADMM is substantially faster than
>    standard ADMM at moderate-to-large $s$ and comparable at small $s$
>    (where the fixed cost of building the screening graph is paid
>    regardless). We use $s$ rather than $(\lambda_1, \lambda_2)$ on
>    the $x$-axis because $s$ is data-adaptive and so invariant to
>    rescalings of the input.

---

### Review · Reviewer_SJLM · 2026-03-01

**Summary Of Contributions:**

Positive:
* I think a computational sufficiency lens for multitask graphical modelling seems interesting to bridge various estimators and connect theory to scalable computation.
* Overall, the paper appears correct in spirit (with some minor issues / typos that should be easily fixable).
* I think the paper tries hard to guide the reader through the technical details; for an average TMLR reader this is highly appreciated.

Problematic / room for improvement:
* From my understanding, constructing the screening graph is not always trivial; and if the graph is dense (depending on data / $(\lambda_1,\lambda_2)$...?) the reduction gives no computational benefit. The paper should better illustrate the range of scenarios where the method helps vs. does not in the multiple matrix scenario.
* I think there are a few manuscript/appendix typos; these need to be corrected or clarified.
* The paper could be more explicit (especially conclusion/discussion) in explaining what is inherited from Vu’s framework vs. what is new (multiview extension + penalty-specific graphs).

**Audience:**

Yes

**Audience Explanation:**

Yes

**Claims And Evidence:**

Yes

**Claims Explanation:**

Aside from a few (likely) presentation/typos, the main claims are broadly supported by a coherent theoretical framework (projection-based computational sufficiency) and minimal empirical evidence showing speedups when the screening graph decomposes into smaller components; however,  the paper could still be clearer about the settings where these gains are not achievable.

**Requested Changes:**

I’d suggest the following concrete things to improve the paper.

* Fix/address the issues/concerns mentioned below.
* Add a small illustrative toy example (e.g., $d=5, K=3$) visualizing the objects $X^{(k)}$, the screening graph/connected components, the mask $W$, and the reduced data $R(X)$.
* Outline best/worst data and $(\lambda_1,\lambda_2)$ regimes (and associated component sizes) to set expectations on when the method provides real and achievable.
* Make sure to clearly state up front that the general computational sufficiency / projection theory (including proof techniques) comes from Vu, and list precisely what is novel here.

Questions/comments:

* Eq. (2.7) (Sparse Covariance example): should the term not be $-\tau \log\det(\theta)$ (a convex log barrier) rather than $+\tau \log\det(\theta)$?
* Appendix §2.3.1 (Group Lasso): the construction for Condition 2 uses
  \[
  X^{(k)}_{ij} = \lambda_1\, U^{(k)}_{ij} + \lambda_2\, V^{(k)}_{ij},
  \]
  but I cannot see how this holds unless $\lambda_2=1$ given the definition of $V$ from the soft-threshold residual in (2.8). Is there a missing division by $\lambda_2$ in the definition of $V$ (or have I overlooked something)?
* Algorithm 1: seems written hastily. The “Require” statement repeats $X^{(1)}$ multiple times, and line 7 uses the wrong variables for the dual update. Please fix and sanity-check the remaining lines and the prox expressions for consistency.

---

> ### Author Response · Authors · 2026-04-13
>
> **Toy example**
>
> We will add a toy example with a diagram to Section 3 to illustrate the different objects.
>
> **Range of scenarios where the method helps**
>
> See General Comment #2. We will add discussion and plots characterizing the regimes where the reduction helps most.
>
> **Novelty attribution**
>
> See General Comment #3. We will make the distinction between Vu (2018) and this paper's contributions explicit in the introduction and discussion.
>
> **Eq. (2.7) sign**
>
> Yes, that is a typo. The sign should be negative, because $-\tau \log\det(\theta)$ serves as a log-determinant barrier that enforces positive definiteness.
>
> **Appendix 2.3.1 Group Lasso**
>
> Thank you for catching this. The decomposition $X^{(k)}\_{ij} = \lambda\_1 U^{(k)}\_{ij} + \lambda\_2 V^{(k)}\_{ij}$ requires that $V^{(k)}\_{ij}$ be normalized by $\lambda\_2$. The current definition gives the soft-threshold residual without this division. We will fix this.
>
> **Algorithm 1**
>
> Thank you for identifying these errors:
> 1. The Require statement has a typo: $X^{(1)}, \dots, X^{(1)}$ should be $X^{(1)}, \dots, X^{(K)}$.
> 2. Line 7 (dual update) uses undefined variables $X^{(k)}\_{t+1}$ and $Y^{(k)}\_{t+1}$. These should be $\Theta^{(k)}\_{t+1}$ and $\beta^{(k)}\_{t+1}$ respectively.
>
> We have identified additional minor errors in the algorithm and will provide a corrected version in the revision.

---

> ### Author Response · Authors · 2026-05-05
> **Changes in response to review**
>
> 1. **Eq. (2.7).** Sign corrected to $-\tau \log\det(\theta)$.
> 2. **Appendix Sec. 2.3.1 (Group Lasso) construction of $V$.** In
>    the two non-zero branches of the formula for $V^{(k)}_{ij}$, the
>    right-hand side has been divided by $\lambda_2$, restoring the
>    penalty-graph condition that the dual entries lie in the unit
>    ball as required.
> 3. **Algorithm 1.** The Require typo, the $\beta$-update subscripts,
>    and the dual-update variables are corrected, and the previously
>    omitted $\rho$ is now in the Require list as "(Step Size)
>    $\rho > 0$."
> 4. **Range of $(\lambda_1, \lambda_2)$ where the reduction helps.**
>    Revised "Dependence on $\lambda_1$ and $\lambda_2$" paragraph in
>    Sec. 3.2.
> 5. **Novelty attribution.** New paragraph at the end of the
>    computational-sufficiency discussion in Sec. 1 and a mirroring
>    sentence in Sec. 6.
> 6. **Toy example with $d=5$, $K=3$.** Added as Figure 1 in Sec. 3.2,
>    showing the input tuple, the screening graph and its connected
>    components, the binary mask, and the reduced data. A second row
>    depicts the lift-back guarantee from Theorem 3.3.

---

### Review · Reviewer_J7zJ · 2026-03-31

**Summary Of Contributions:**

- Contribution: The paper applies computational sufficiency in defining a universal screening rule for the joint estimation of multiple matrices. It provides a theoretical guarantee for the reduction on the family of estimators under certain conditions, and also provides empirical evidence to support computational reduction in utilizing ADMM for sparse PCA/glasso estimators.

- Strengths:
  - The paper provides a unifying framework for a family of estimators for joint matrix estimation regardless of algorithms.
  - The paper offers demonstrative examples for a penalized loss function.

- Weakness:
  - The paper does not address possible limitations. For example, the theorem limits to estimators with diagonal sign conjugation invariant generators, and the scope of generality is not discussed.
  - The experiment is only concerned with one algorithm, and it is not convincing to support the computational reduction independent of algorithms.
  - The paper needs more careful notation.

**Audience:**

Yes

**Audience Explanation:**

Unifying a group of estimators depending solely on data is an interesting direction to improve algorithm-agnostic analysis and improvement.

**Claims And Evidence:**

Yes

**Claims Explanation:**

- Limitation: For example, the theorem limits to estimators with diagonal sign conjugation invariant generators, and the scope of generality is not discussed.
- Experiment: More experiments with different algorithm and estimators should be applied to better support the claim.
- Notation: For example $\lambda_2$ used in sec. 1.1 was not defined until eq (2.2) and its range is not well defined. $\tau$ in Definition 3.1 is undefined.

**Requested Changes:**

- Add more experiments to support the algorithm-agnostic claim.
- Address limitations of the proposed unifying framework.

---

> ### Author Response · Authors · 2026-04-13
>
> **Limitations and scope of generality**
>
> Diagonal sign conjugation invariance (Condition 1) captures the symmetry that arises whenever variable orientation is arbitrary (e.g., flipping a feature's sign). It is satisfied by the loss functions for all the core applications we discuss (precision matrices, PCA, covariance estimation) and defines a class strictly more general than orthogonally invariant functions. It is also the maximal symmetry group that preserves sparsity patterns. We will add a discussion explicitly characterizing the scope of these generators.
>
> **Experiments with different algorithms**
>
> See General Comment #1. We will address this with both a theoretical corollary and additional experiments using a proximal gradient solver.
>
> **Notation**
>
> - **$\lambda_2$ in Section 1:** $\lambda_2$ is mentioned informally in the contributions summary (Section 1) before its formal definition in Eq. (2.2). We will either add a forward reference or briefly define it at first use.
> - **$\mathcal{X}$ and $\mathcal{T}$ in Definition 3.1:** We will add definitions for these. They are the input and output spaces of the estimators; the estimator is set-valued to accommodate the case of multiple optima.

---

> ### Author Response · Authors · 2026-05-05
> **Changes in response to review**
>
> 1. **Limitations of the diagonal sign-conjugation invariance
>    condition.** New paragraph "Scope of the invariance condition" in
>    Sec. 6.
> 2. **Algorithm-agnostic claim.** Theoretical: new **Corollary 3.4**
>    in Sec. 3.2 (proof in the appendix). Empirical: new Figure 3 in
>    Sec. 5.1 (proximal-gradient solver), with full setup in the
>    appendix.
> 3. **Notation.**
>    - $\lambda_2$ in Sec. 1 is now glossed inline at first mention with a
>      pointer to Eq. (2.2).
>    - $\tau$ in the sparse-covariance generator is defined immediately
>      following Eq. (2.7).
>    - $\mathcal{X}$ and $\mathcal{T}$ in Definition 3.1 are defined as
>      input and parameter spaces; both equal $(\mathbb{R}^{d \times d})^K$
>      for our family.

---

### Author Response · Authors · 2026-04-13
**General Comment (to all reviewers)**

We thank all three reviewers for their careful reading and constructive feedback. We address each reviewer's comments individually below and summarize the main changes:

1. **Algorithm-agnostic claim.** Two reviewers noted that our experiments only use ADMM. We will add a corollary showing that the block-diagonal structure of the reduced problem is inherited by the gradient, subgradient, and proximal operator of the objective, so *any* first-order method automatically decomposes into independent subproblems over blocks. The computational benefit follows from the reduction itself, not the choice of solver. We will also add experiments with a proximal gradient solver to demonstrate this empirically.

2. **When does the reduction help?** First, a clarification: our experiments measure total runtime to compute a full regularization path (100 penalty values), which is the standard workflow when tuning parameters by cross-validation. The current Figure 1 caption does not make this clear, and we will revise it. For a single penalty value, the reduction is most effective when regularization is strong (large $\lambda_1, \lambda_2$). This is the regime of greatest practical interest for high-dimensional problems. When regularization is weak, the screening graph is dense and the reduction provides little benefit, but the original problem is also relatively easy to solve. We will add discussion and plots characterizing these regimes, including execution time vs. sparsity of the thresholded data as suggested by Reviewer mE5f.

3. **Novelty attribution.** We will clarify the relationship to Vu (2018), which provides an abstract group-theoretic framework but does not work out any of the cases in this manuscript: that work is limited to the single matrix case without fusion penalties. Our contributions are: (a) extending the framework to joint multiple matrix estimation, (b) deriving penalty-specific screening graphs for group lasso and generalized lasso fusion penalties, and (c) the resulting universal screening rule and algorithms. We will make this distinction explicit in the introduction and discussion.

4. **Presentation improvements.** We will fix the notation issues, Algorithm 1 errors, and typos identified by the reviewers.

---

> ### Comment · Reviewer_mE5f · 2026-04-22
> **Main changes**
>
> Have the authors posted an updated manuscript with these changes?

---

> > ### Author Response · Authors · 2026-04-22
> >
> > Thank you for the reminder. We are preparing the revised manuscript.

---

### Comment · Action_Editor_8VTp · 2026-04-28
**Delay of revision**

Dear Reviewers,

Please note that I have approved a delay in the submission of the revision by the authors to May 5 (instead of the original April 28). Please take this into account before providing any further feedback.

Regards,
Dan

---

### Author Response · Authors · 2026-05-05
**Revised manuscript posted**

We have posted a revision on OpenReview. A detailed change list is
in the *Changes Since Last Submission* field of the revision.
Per-reviewer replies addressing each reviewer's individual points
are posted under each review.

---

### Decision · Action_Editor_8VTp · 2026-06-21

**Recommendation:** Accept as is

**Audience:**

Yes

**Audience Explanation:**

Both reviewers clearly indicate that this work is interesting and relevant.

**Claims And Evidence:**

Yes

**Claims Explanation:**

The paper contains both concrete theoretical contributions and convincing numerical results. Both reviewers are satisfied with the presentation in the last revision.

---

> ### Author Response · Authors · 2026-07-20
> **Camera-Ready Version Submitted**
>
> We have submitted the camera-ready version of our paper. Thank you for your guidance, support, and constructive comments throughout the review process.